# NEDDylation promotes nuclear protein aggregation and protects the Ubiquitin Proteasome System upon proteotoxic stress

Chantal M. Maghames[1], Sofia Lobato-Gil[1,2], Aurelien Perrin[1], Helene Trauchessec[1], Manuel S. Rodriguez[2], Serge Urbach[3], Philippe Marin[3] & Dimitris P. Xirodimas [1]

Spatial management of stress-induced protein aggregation is an integral part of the proteostasis network. Protein modification by the ubiquitin-like molecule NEDD8 increases upon proteotoxic stress and it is characterised by the formation of hybrid NEDD8/ubiquitin conjugates. However, the biological significance of this response is unclear. Combination of quantitative proteomics with biological analysis shows that, during proteotoxic stress, NEDDylation promotes nuclear protein aggregation, including ribosomal proteins as a major group. This correlates with protection of the nuclear Ubiquitin Proteasome System from stress-induced dysfunction. Correspondingly, we show that NEDD8 compromises ubiquitination and prevents targeting and processing of substrates by the proteasome. Moreover, we identify HUWE1 as a key E3-ligase that is specifically required for NEDDylation during proteotoxic stress. The study reveals a specific role for NEDD8 in nuclear protein aggregation upon stress and is consistent with the concept that transient aggregate formation is part of a defence mechanism against proteotoxicity.

[1] CRBM, CNRS, Univ. Montpellier, UMR5237, Montpellier Cedex 5, 34090, France. [2] ITAV CNRS USR3505, IPBS-UPS, 1 Place Pierre Potier, Oncopole Entrance B, 31106 Toulouse, France. [3] Institute of Functional Genomics, 141 rue de la Cardonille, 34094 Montpellier Cedex 5, France. Correspondence and requests for materials should be addressed to D.P.X. (email: dimitris.xirodimas@crbm.cnrs.fr)

Maintaining a proper and functioning proteome is challenging for cells given the variety of proteotoxic stresses that constantly cause protein damage[1,2]. Aberrant proteins tend to form aggregates that have been linked to many neurodegenerative diseases (Alzheimer's disease, Parkinson's disease), cancer and aging[3–7]. To prevent aggregation, a protein quality control (PQC) network ensures protein repair and refolding by molecular chaperones or clearance of terminally damaged proteins by lysosomal and proteasomal degradation[8,9]. While PQC systems in the cytosol have been well-studied[8], knowledge on the nuclear PQC is still limited, especially in mammalian cells. The Ubiquitin Proteasome System (UPS) is an essential PQC pathway, as it eliminates misfolded proteins prone to aggregation. Degradation by the 26S proteasome depends on the covalent modification of substrates with ubiquitin. Upon activation by the E1-activating enzymes UBA1 or UBA6, ubiquitin is transferred to E2-conjugating enzymes before the modification mainly of lysine residues on target proteins through the action of E3 ligases[10]. Upon ubiquitination, proteins are recognised by shuttle factors that ensure substrate delivery to the 26S proteasome. Receptors on the 19S lid subunit of the proteasome allow recognition of ubiquitinated proteins followed by ATP-dependent protein unfolding and de-ubiquitination, before translocation of the substrate into the catalytic 20S core for degradation[11]. Each of the above steps is independently and finely regulated and collectively they define the rate of substrate degradation[12]. Proteasomes exist both in the cytoplasm and nucleus of mammalian cells, but their relative abundance and activity depend on the cell type and growing conditions[13]. Dysfunction of the UPS is often related to the appearance of protein aggregates in several neurodegenerative diseases[14–16]. However, it is not clear whether formation of aggregates is always a cause of UPS toxicity[14–17] or it can be part of a defence mechanism against proteotoxicity[18–21].

While ubiquitin is considered as a main effector in the PQC network, members of the ubiquitin-like molecules family are emerging as new and important regulators of protein homoeostasis[22,23]. NEDD8 is the closest homologue to ubiquitin with 60% sequence identity and it is conjugated to proteins in a three-step process, similarly to ubiquitin. Nevertheless, NEDD8 employs its own specific enzymes, a heterodimer NEDD8 E1-activating enzyme (NAE) and NEDD8 E2-conjugating enzymes, UBE2M (UBC12) and UBE2F[24,25]. The NAE-dependent functions of NEDD8 have been well characterised, mainly in homoeostatic conditions, through NEDDylation of the cullin family of proteins and stimulation of the Cullin-Ring-Ligases (CRLs) activity, but also through NEDDylation of non-cullin substrates[24,25]. Early studies showed that in several neurodegenerative disorders NEDD8 colocalises with ubiquitin and proteasome components in protein inclusions, suggesting a role for NEDD8 in protein aggregation[26–28]. More recently, a role for the NEDD8 interacting protein NUB1 was proposed in the regulation of tau aggregation in Alzheimer's disease[29] and mutant Huntington toxicity[30]. In addition, a direct link between the NEDD8 pathway and proteotoxic stress was also revealed. Exposure of cells to heat shock, proteasome inhibitors or oxidative stress increases protein NEDDylation. However, the stress-induced NEDD8 conjugation depends on the ubiquitin E1-activating enzyme UBA1 and not on NAE. This was quite surprising given the specificity of E1 enzymes in activating their cognate molecules[31]. A key characteristic of the NEDD8 response to stress is the formation of hybrid NEDD8/ubiquitin conjugates[32]. While the above studies indicate a link between NEDD8 and proteotoxicity, the biological significance of the NEDD8 response to proteotoxic stress remains unclear. We report that NEDD8 promotes nuclear protein aggregation during proteotoxic stress and

protects the nuclear UPS from stress-induced dysfunction. Mechanistically, NEDD8 compromises substrate ubiquitination and prevents their processing/degradation by the proteasome. Importantly, these effects are independent of NEDD8 activation by NAE and of the CRL function. This study identifies NEDDylation as a regulatory pathway of protein aggregation and UPS function in the nucleus during proteotoxic stress and is consistent with the concept that transient protein aggregation is a defence mechanism against proteotoxicity.

## Results

**Characterisation of the NEDD8 response to proteotoxic stress.** In addition to pre-existing proteins, newly synthesised polypeptides are particularly vulnerable and damaged under proteotoxic stress[33]. Treatment of cells with the translational inhibitor Cycloheximide (CHX) prevented the increase in protein NEDDylation upon HS, indicating that the NEDD8 response to proteotoxic stress depends on protein synthesis (Fig. 1a). To determine if the stress-induced NEDDylation depends, at least in part, on damage of newly synthesised proteins we used puromycin, a protein synthesis inhibitor known to induce shortening and damage of newly synthesised proteins at low doses (5–20 μg/ml) and to completely block protein synthesis at high doses (100 μg/ml)[34]. Low concentrations of puromycin induced protein NEDDylation in the absence of any additional stress and showed an additive effect on NEDD8 accumulation when combined with HS. In contrast, when used at a high dose, puromycin reduced the NEDD8 response to HS similarly to the CHX treatment (Fig. 1b). Short-term inhibition of HSP70, which is expected to increase the production of misfolded newly synthesised proteins[35], also induced NEDDylation, consistent with the data obtained with puromycin (Supplementary Figure 1A). Pulse labelling experiments, where newly synthesised $^{35}$S-labelled NEDD8 was isolated by immunoprecipitation, showed that the increase in NEDDylation upon stress is not due to an increase in NEDD8 synthesis (Supplementary Figure 1B).

To determine the state of stress-induced NEDDylated proteins, we performed a subcellular fractionation in untreated and HS cells, separating soluble (cytosol, nucleoplasm) from insoluble (pellet) fractions. NEDDylated substrates showed a transition from the cytosolic soluble into the insoluble pellet fraction during HS, indicating aggregation of the modified substrates (Fig. 1c). A well-characterised effect of proteotoxic stress is to induce the formation of cytoplasmic protein aggregates known as aggresomes[36]. However, by immunofluorescence analysis we found that NEDD8 accumulated only in the nucleus upon HS and colocalised with ubiquitin (Fig. 1d). Interestingly, when cells were subjected to a severe proteotoxic stress (overnight MG132 treatment), we detected a reorganisation of the nuclear NEDD8/ubiquitin aggregates (Fig. 1e, Supplementary Figure 2A). 3D reconstruction showed that the observed structures are a sphere and that NEDD8 and ubiquitin colocalise on its surface (Fig. 1e).

Stress-induced NEDDylation is reported to depend on the activation of NEDD8 by the ubiquitin E1 enzyme UBA1 and not by NAE. Indeed, no effect on HS-induced NEDD8/ubiquitin-stained nuclear aggregates was observed upon inhibition of NAE with the specific inhibitor MLN4924 (Fig. 1f). In contrast, treatment of cells with the specific UBA1 inhibitor MLN7243 caused a dramatic decrease in NEDD8/ubiquitin-stained aggregates (Fig. 1f). A dramatic decrease in NEDD8 but not ubiquitin staining in nuclear aggregates was observed upon NEDD8 knockdown by siRNA, showing the antibody specificity (Fig. 1f). Similar results were obtained upon severe proteotoxic stress with MG132 (Supplementary Figure 2B). These data suggest that

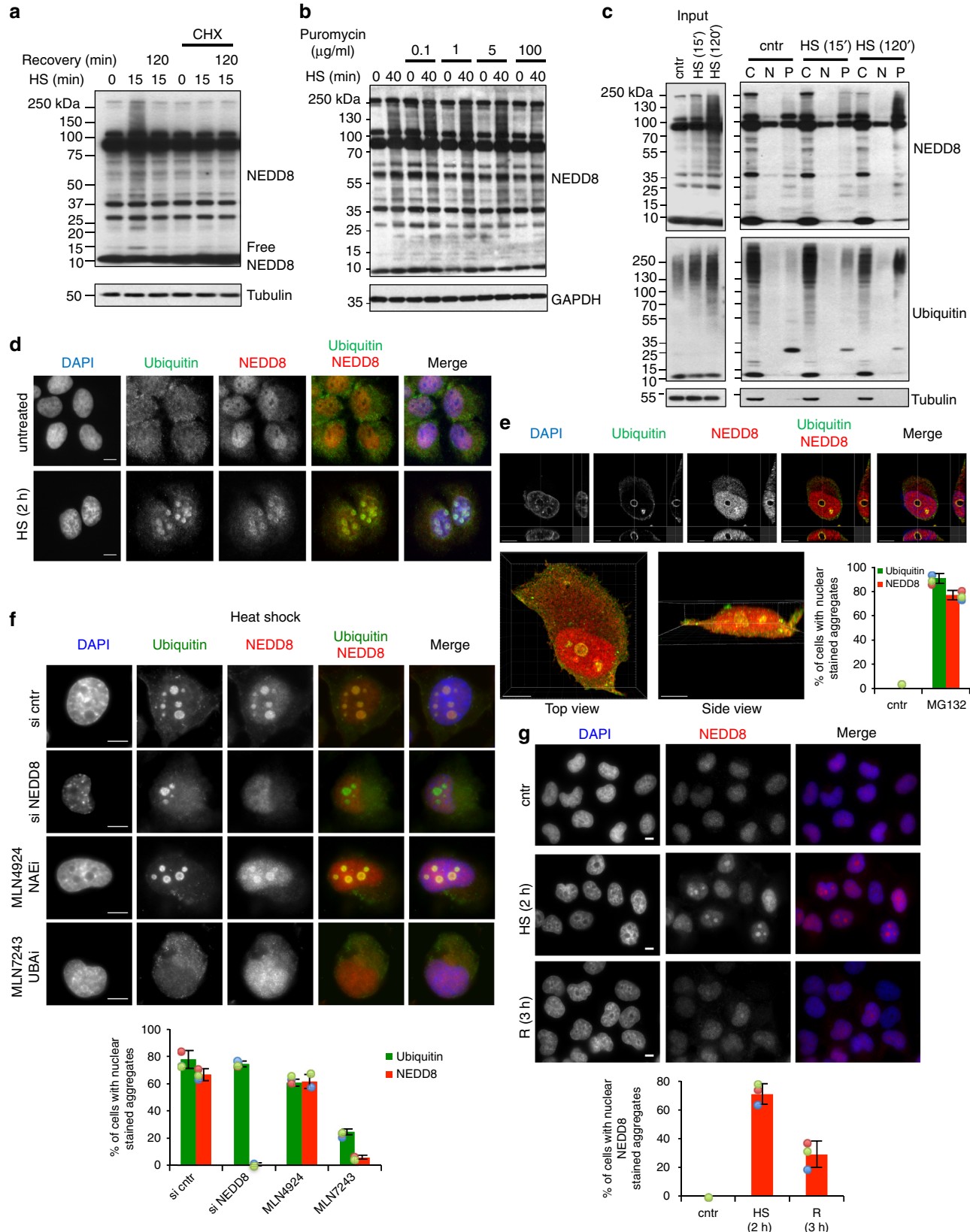

the formation of stress-induced NEDD8 nuclear aggregates depends on the activation of NEDD8 by UBA1. The HS-induced NEDD8/ubiquitin aggregates are transient and disappear when cells are allowed to recover at 37 °C (Fig. 1g, Supplementary

Figure 2C). The combination of the above data indicates that HS induces a progressive accumulation of transient and reversible NEDD8 protein aggregates in the nucleus through NEDD8 activation by UBA1.

**Fig. 1** NEDD8 targets newly synthesised proteins and accumulates in nuclear aggregates upon proteotoxic stress. **a** U2OS cells were pretreated or not with Cycloheximide (CHX) at 100 µg/ml for 90 min, and then treated as indicated. After heat shock at 43 °C, cells were recovered at 37 °C for 2 h. **b** U2OS cells were pretreated or not with puromycin at the indicated concentrations for 30 min then heat-shocked. Cell extracts were analysed by western blotting. **c** U2OS cells were either untreated or heat-shocked at 43 °C for the indicated time. After HS, cells were subjected to subcellular fractionation as described in Methods (C, Cytoplasm; N, Nucleoplasm; P, Pellet) and fractions were analysed by western blotting. Tubulin was used as fractionation marker. **d** U2OS cells were either untreated or heat-shocked before staining for NEDD8 (red) and ubiquitin (green). Nuclei were stained with DAPI (blue). **e** Similar as in **d**, but cells were pretreated with MG132 (5 µM-overnight-ON). 3D rendering was performed as described in Methods. Top panel represents single plane images in different axis. Bottom images are the 3D reconstructions for the top and side view. Scale bar: 10 µm. Bottom right graph represents the % of cells with nuclear aggregates stained with NEDD8 or ubiquitin. Approximately 100 cells were counted per condition. **f** siRNA transfection in H1299 cells was performed as indicated for a total of 36 h. MLN4924 and MLN7243 were used at 500 nM for 15 and 5 h respectively before heat shock. Staining and quantitation of the experiment (bottom graph) were performed as above. **g** H1299 cells were heat-shocked as before and were allowed to recover at 37 °C as indicated. After fixation, cells were stained for NEDD8 (red) and DAPI (blue). Bottom graph represents quantitation of the experiment in (**g**), performed as above. For all bar graphs, values represent the average of three independent experiments ± standard deviation (SD). Scale bars, 10 µm

**NEDDylation promotes nuclear aggregates formation and protects nuclear UPS.** Based on the ubiquitin staining, the immunofluorescence analysis indicates that the formation of nuclear aggregates upon proteotoxic stress is not dramatically affected upon NEDD8 knockdown (Fig. 1f, Supplementary Figure 2B). To determine whether instead the composition of nuclear aggregates is controlled by NEDD8, we followed a SILAC mass spectrometry-based proteomic approach. We first determined the composition of protein aggregates induced by HS. U2OS cells were labelled with light (K0R0) or heavy (K8R10) medium and either left at 37 °C (light) or heat-shocked (heavy) at 43 °C for 2 h. Cells were mixed 1:1 ratio before subcellular fractionation, isolation of the insoluble pellet fraction and mass spectrometry analysis (Fig. 2a). Peptides for approximately 1500 proteins were quantified in two replicate experiments (Supplementary Data 1). Consistent with the western blot and immunofluorescence analysis, ubiquitin and NEDD8 increased in abundance in the formed insoluble inclusions upon HS (Fig. 2b). Increased abundance was also observed for SUMO-1, 2 in accordance with the established role of SUMOylation in the HS response (Fig. 2b) and the emerging concept for the cross-talk between ubiquitin and Ubls upon stress conditions[22,37,38]. A striking observation was that among proteins with increased abundance in the aggregates, RNA transport and ribosomal proteins (RPs) were the two major groups, accounting for approximately 40% of the total number (Fig. 2b, c).

Having determined the composition of the HS-induced aggregates we then assessed the potential role of NEDD8 in the stress-induced nuclear protein aggregation. Based on previous studies[39], we found that short-term (36−48 h) knockdown of NEDD8, while it fully blocks the stress-induced NEDD8 response, it does not have a major impact on cullin NEDDylation, a key target of the NAE-dependent NEDD8 conjugation (Fig. 2d)[24,25]. Importantly, under these conditions the levels of well-established targets of CRLs (p21, CDT1) remain unaffected, indicating that the small decrease observed in cullin NEDDylation has no significant effect on the CRL function (Supplementary Figure 3). While these observations suggest that NEDD8 knockdown mainly impacts on stress-induced NEDDylation, defects due to inhibition of NAE-dependent NEDDylation cannot be excluded. To discriminate these effects we performed two SILAC experiments: We determined the effect of NEDD8 knockdown on the HS-induced aggregate composition and in parallel we performed a similar experiment using instead the NAE inhibitor MLN4924 (Fig. 2e, Supplementary Figure 4A, B). As shown (Fig. 2d, Supplementary Figure 3), MLN4924 has no effect on stress-induced NEDDylation but almost completely blocks NAE-dependent NEDDylation, including cullin modification and CRL function. We reasoned that the comparison of the two experiments should eliminate any effects due to inhibition of

NAE-dependent NEDDylation and indicate the role of stress-induced NEDDylation on protein aggregation.

Comparison of the two SILAC experiments shows that NEDD8 knockdown affects the composition of the HS-induced aggregates (Fig. 2f, Supplementary Figure 4A, B). Both reduced and increased aggregation was observed (Fig. 2f, g, Supplementary Figure 4C), but clearly the most profound effect of NEDD8 knockdown is to reduce protein aggregation. In particular, a large number (30 IDs) of RPs (Fig. 2f, g) was identified as the most enriched group of proteins with reduced aggregation. Importantly, these effects were specific for NEDD8 knockdown and not observed upon inhibition of the NAE-dependent NEDD8 pathway and CRL function by MLN4924 (Fig. 2f, g, Supplementary Figure 4B). The proteomic analysis indicates that proteotoxic stress-induced NEDDylation mainly promotes aggregation.

The mutual functional relationship between protein aggregation and UPS function is well-established. However, it is still controversial whether stress-induced protein aggregation is detrimental for UPS function or rather a defence mechanism to protect UPS from the increase load of misfolded proteins during stress[14–16]. We therefore determined the role of NEDDylation in UPS function during the HS response. We used HEK293 cells stably expressing GFPu, a fusion of GFP with a peptide that targets GFP for ubiquitin-dependent proteasomal degradation. By monitoring the protein levels of GFPu, the UPS activity can be assessed[40]. We used GFPu constructs fused either to a nuclear export (NES/cytoplasmic) or nuclear import (NLS/nuclear) signal, thus allowing the assessment of the cytoplasmic or nuclear UPS activity[18]. While HS had no effect on NES-GFPu, a progressive increase in NLS-GFPu was observed, indicating that HS specifically compromises the UPS in the nucleus (Fig. 3a, b). This could be due to the reduced UPS activity in the nucleus compared to the cytoplasm or due to the lack of auxiliary proteolytic machineries such as the autophagy pathway[13,35]. Importantly, once stress is alleviated, nuclear UPS is recovered (Fig. 3b). Inhibition of NEDDylation by NEDD8 knockdown further compromised the nuclear UPS activity with no effect on cytoplasmic UPS. Importantly, NEDD8 knockdown reduced UPS activity only during the HS response and not in unstressed conditions. No significant effects on UPS function were observed upon inhibition of the NAE-dependent NEDDylation by MLN4924 (Fig. 3c, d, Supplementary Figure 5A). Monitoring the expression of NLS-GFPu mRNA and rates of protein synthesis shows that the observed effects of NEDD8 knockdown on NLS-GFPu levels are not due to transcriptional/protein synthesis defects (Supplementary Figure 5B, C). Ectopic expression of NEDD8 resistant to siRNA restores the NEDD8 response to stress and rescues the effect of NEDD8 knockdown on nuclear UPS activity (Fig. 3e). The above-described defects in the UPS upon inhibition of NEDDylation could be explained by a

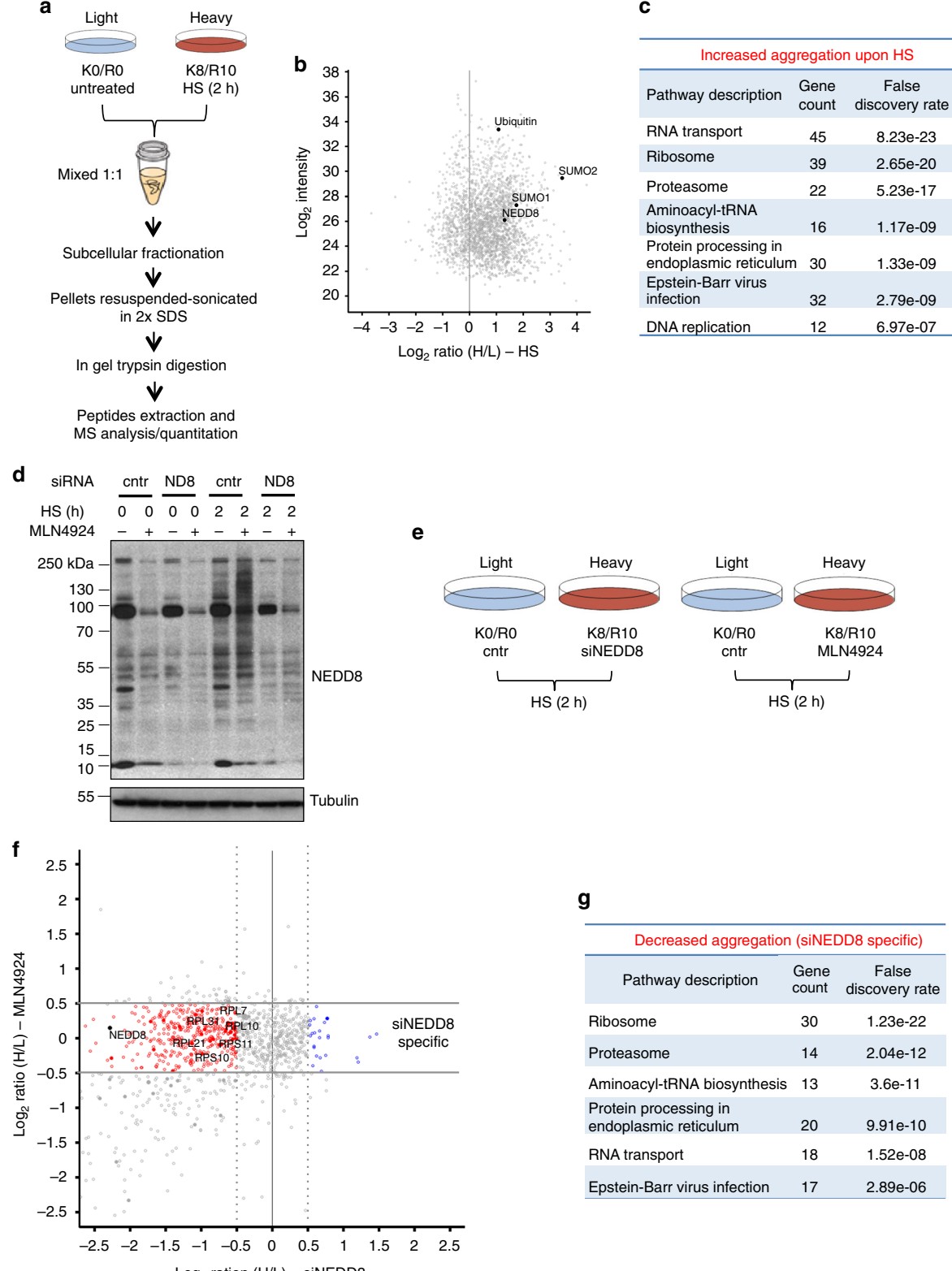

reduction in proteasome activity. However, experiments in cytoplasmic and nuclear extracts showed that neither HS nor NEDD8 knockdown reduced the intrinsic activity of the proteasome (Fig. 3f). In contrast, a small but reproducible increase in nuclear proteasome activity was observed upon NEDD8 knockdown (Fig. 3f). The data suggest that NEDDylation

specifically protects the nuclear UPS during the HS response through regulation of protein aggregation.

**Proteotoxic stress promotes NEDDylation of RPs.** To better understand the role of NEDDylation in UPS function regulation

**Fig. 2** Quantitative proteomics to define the role of NEDD8 in HS-induced protein aggregation. **a** SILAC proteomics strategy to determine the composition of the HS-induced aggregates. **b** Scatter plot of the mean of two replicate experiments for the HS SILAC. **c** Table for the group of proteins with increased aggregation upon HS. The top seven groups are presented, including the number of identified proteins with H/L SILAC ratio >1 and the false discovery rate (FDR). **d** U2OS cells were transfected with control or NEDD8 siRNA or treated with MLN4924 (200 nM) for 48 h, were heat-shocked or not as indicated. Cell extracts were used for western blotting for the indicated proteins. **e** SILAC strategy to determine the effect of either NEDD8 knockdown or MLN4924 on the composition of HS-induced aggregates. **f** Scatter plot comparing the siNEDD8 and MLN4924 SILAC experiments performed in **e**. Data are the mean of two replicate experiments for each condition. In red and blue are proteins that are either less or more aggregated respectively during HS, specifically upon NEDD8 knockdown but not upon MLN4924 treatment. **g** Table for the group of proteins with decreased aggregation upon HS specifically observed upon NEDD8 knockdown

upon proteotoxic stress, our studies focussed on the characterisation of model substrates that aggregate upon HS in an NEDD8-dependent but NAE-independent manner. The proteomic analysis identified RPs as a major group of proteins that fulfil the above criterion (Fig. 2f, g). In addition, RPs have been reported as NEDD8 substrates and more recently implicated in the proteotoxic stress response[41,42]. We initially determined whether RPs found in aggregates upon proteotoxic stress are substrates of NEDDylation. HCT116 cells stably expressing His$_6$-NEDD8 were heat-shocked or treated with MG132, then subjected to a subcellular fractionation. His$_6$-tagged NEDDylated proteins isolated from solubilised aggregates were blotted against RPL7, an RP identified in the proteomics analysis and used as model substrate. The data show that proteotoxic stress increases NEDDylation of RPL7, which is found in the insoluble pellet (Fig. 4a). The use of the specific inhibitors of NAE (MLN4924) and UBA1 (MLN7243) shows that stress-induced NEDDylation of RPL7 depends on UBA1 and not on NAE (Fig. 4b). The UBA1-dependent NEDDylation is characterised by the simultaneous modification of the substrate both with NEDD8 and ubiquitin, including the formation of hybrid NEDD8/ubiquitin chains[32]. To determine the modification of RPL7 both with NEDD8 and ubiquitin, we followed a double purification approach. Based on previous studies, we used the UBA domain of DSK2 protein, which in addition to polyubiquitin it also interacts with hybrid NEDD8/ubiquitin conjugates[32]. Extracts of His$_6$-NEDD8-U2OS cells treated with MG132 were used for a GST-UBA pull-down, before eluates were applied in a sequential His$_6$-NEDD8 purification under denaturing conditions. Western blot analysis shows the isolation of high molecular weight conjugates of RPL7, which indicates the simultaneous modification of RPL7 with NEDD8/ubiquitin possibly by hybrid chains (Supplementary Figure 6). The detection of mainly high molecular weight RPL7 conjugates is most likely due to the preferential isolation and enrichment of poly-modified conjugates by the UBA domains.

Furthermore, immunofluorescence analysis shows that severe proteotoxic stress (MG132-ON) causes the accumulation of RPL7 within the previously characterised nuclear structures stained with NEDD8 and ubiquitin (Fig. 5a, b). Similar results were obtained for RPL11, also identified in the proteomic analysis (Supplementary Figure 7). As RPs are key components of the nucleolus, we performed costaining of NEDD8 and ubiquitin with other nucleolar proteins, fibrillarin and nucleolin. We found no colocalisation, suggesting that the observed NEDD8/ubiquitin nuclear structures are distinct of the nucleolus, rather a stress-induced nucleolar-related structure composed of RPs (Fig. 5c, d).

To further confirm the role of stress-induced NEDDylation in RPs aggregation we applied an overexpression approach, as previous studies showed that ectopically overexpressed NEDD8 is activated by the ubiquitin UBA1 E1 enzyme. These conditions mirror the stress-induced NEDDylation observed under endogenous levels of NEDD8 expression upon proteotoxic stress[32,43]. Immunofluorescence analysis showed that overexpressed NEDD8 is found in nuclear inclusions in unstressed conditions, similar to

the ones observed with endogenous NEDD8 upon HS (Fig. 6a). Importantly, under these conditions NEDD8 overexpression promotes the aggregation of RPL7, although to a smaller extent compared to heat shock (Fig. 6b). In contrast, nucleolin, which was shown in the proteomic analysis (Supplementary Data 1) to aggregate upon HS but independently of NEDD8, was not affected by NEDD8 overexpression (Fig. 6b). In addition, ubiquitin overexpression does not lead to substrate aggregation in the absence of stress (Supplementary Figure 8). Overexpression of the NUB1 (NEDD8 Ultimate Buster 1) protein and its close homologue NUB1L was shown to dramatically repress stress-induced NEDDylation in several cell lines[44,45]. We used NUB1 overexpression as an additional tool to determine the role of stress-induced NEDDylation in protein aggregation. Overexpression of NUB1 reduced both HS-induced NEDDylation and RPL7 aggregation (Fig. 6c). The combination of the above data is consistent with the proteomic analysis, indicating a direct role of stress-induced NEDDylation in promoting nuclear aggregation of RPs upon proteotoxic stress.

**HUWE1 is specifically required for stress-induced NEDDylation.** While it is established that upon proteotoxic stress NEDD8 is activated by the ubiquitin E1 enzyme leading to hybrid NEDD8/ubiquitin conjugates, the identity of conjugating enzymes that promote NEDDylation during proteotoxic stress remains unknown. Recent studies identified Tom1/HUWE1 as an E3-ligase that is required for RPs ubiquitination upon proteotoxic stress[46]. Our proteomic studies identified HUWE1 as one of the proteins enriched in HS-induced aggregates (Fig. 7a). These data were confirmed by immunofluorescence, which showed that proteotoxic stress causes the appearance of HUWE1 in nuclear aggregates, reminiscent to those observed for NEDD8/ubiquitin (Fig. 7b). We therefore tested the role of HUWE1 in stress-induced NEDDylation. Knockdown of HUWE1 significantly reduced the NEDD8 response to proteotoxic stress, including RPL7 NEDDylation, with no significant effect either on cullin NEDDylation (Supplementary Figure 9) or on the global ubiquitin response to proteotoxic stress (Fig. 7c, d). Consistent with the above, immunofluorescence analysis showed that HUWE1 knockdown caused a dramatic decrease in NEDD8 but not ubiquitin-stained nuclear aggregates upon proteotoxic stress (Fig. 7e, f). The presence of multiple E3-ligases could potentially compensate for the loss of HUWE1, consistent with the small effect of HUWE1 knockdown on global ubiquitination. Thus, the dramatic decrease in stress-induced NEDDylation, including RPL7 as a substrate, observed upon HUWE1 knockdown, suggests that HUWE1 is the major E3-ligase that promotes NEDDylation specifically during proteotoxic stress (Fig. 7c–f).

**NEDDylation compromises proteasomal substrate processing.** The experiments in Fig. 3 suggested that inhibition of NEDDylation enhances UPS dysfunction during stress. However, this is not due to a decrease in the intrinsic activity of the proteasome

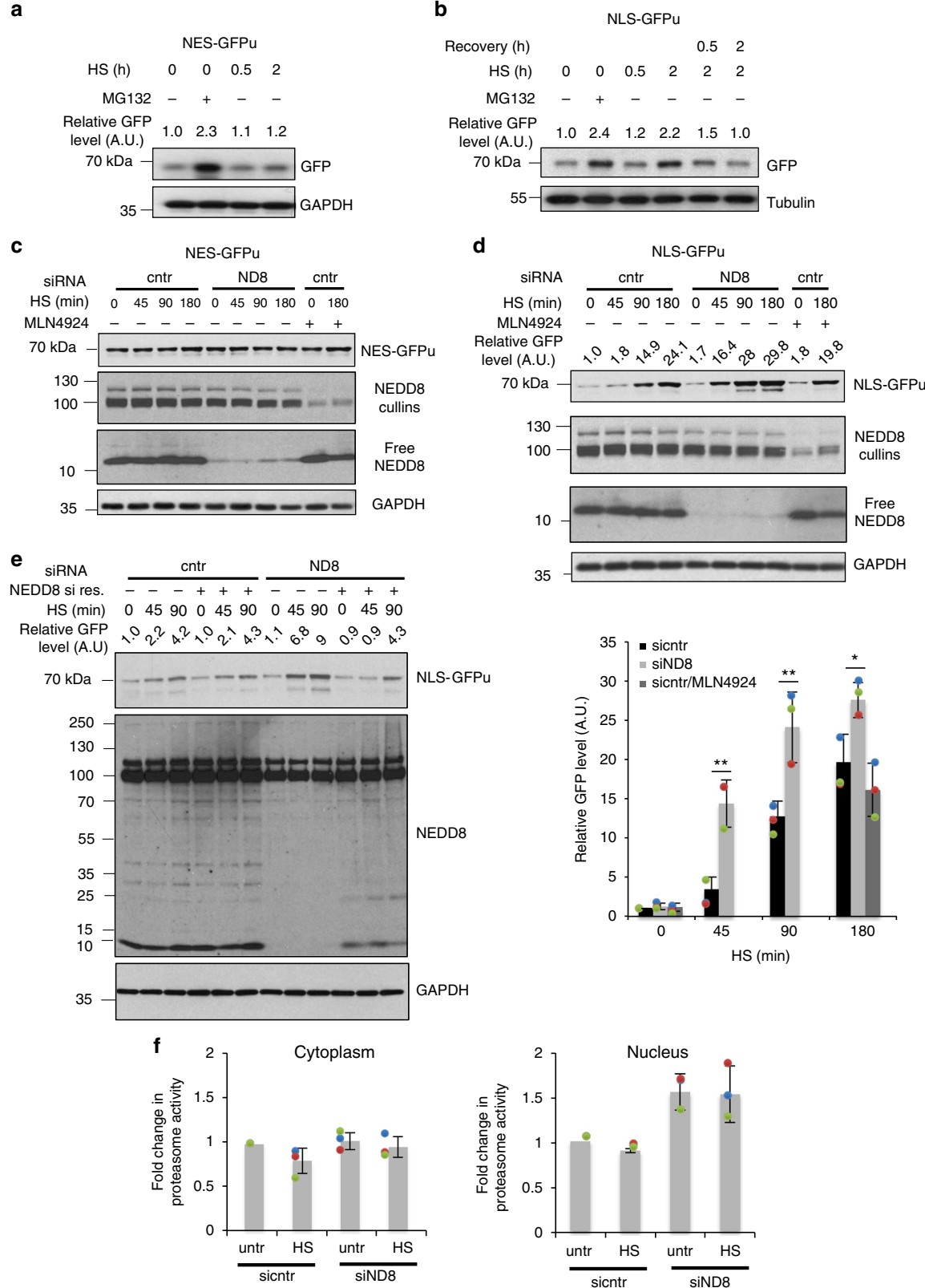

(Fig. 3f), leading us to the hypothesis that NEDDylation could regulate additional steps within the UPS. These include substrate modification and delivery to the proteasome. We used RPL7 as a model substrate to test the above hypothesis. Stress-induced NEDDylation is characterised by the simultaneous modification of substrates with NEDD8 and ubiquitin (Supplementary

Figure 6), raising the possibility that NEDD8 can compromise substrate ubiquitination, the natural targeting signal to the proteasome. Knockdown of NEDD8 increased RPL7 ubiquitination consistent with the idea that NEDD8 conjugation compromises substrate ubiquitination (Fig. 8a). Similar results were obtained upon NUB1 overexpression, which blocks

**Fig. 3** NEDD8 protects the UPS from severe impairment during HS. HEK293 cells stably expressing NES-GFPu (**a**) or NLS-GFPu (**b**) were either untreated or heat-shocked at 43 °C. Recovery was performed at 37 °C. MG132 (25 μM for 5 h) was used as positive control for UPS inhibition. Cell extracts were blotted against the indicated proteins. **c**, **d** HEK293 cells stably expressing NES-GFPu (**c**) or NLS-GFPu (**d**) were transfected with nontarget (cntr) or NEDD8 (ND8) siRNAs, or treated with MLN4924 (200 nM). After 48 h cells were heat-shocked for the indicated periods and cell extracts were blotted against the indicated proteins. Bottom right graph: Quantitation of three independent experiments performed as in **d** ± SD. Two-tailed unpaired Student's $t$ test, *$p < 0.05$, **$p < 0.01$. **e** Similar experiment as above using the HEK293 NLS-GFPu cells. Twenty-four hours after siRNA transfection, cells were transfected either with pcDNA3-expressing siRNA-resistant NEDD8 or empty vector. **f** U2OS cells transfected with control or NEDD8 siRNA were either untreated or heat-shocked for 30 min. Cells were harvested and proteasome activity in cytoplasmic or nuclear extracts was measured as described in Methods. Graphs represent the average of the fold change in proteasome activity ± SD from three independent experiments

RPL7-NEDDylation and promotes RPL7-ubiquitination (Supplementary Figure 10A, B). Importantly, the observed increase in RPL7 ubiquitination upon NEDD8 knockdown depends on HUWE1, further confirming the specific role of HUWE1 in RP modification[46], possibly as a dual E3 using both ubiquitin and NEDD8 for substrate modification (Fig. 8b).

Under similar conditions we monitored the interaction of RPL7 with the proteasome. We used HEK293 cells stably expressing a His-biotinylatable-tagged proteasome subunit (RPN11-HTBH), which is a well-established system to isolate proteasomes[47]. Proteasomes were purified with streptavidin beads as described in Methods and the binding to HA-RPL7 was monitored by western blotting. Exposure of cells to HS induced the RPL7−proteasome interaction, which was further stimulated upon NEDD8 knockdown (Fig. 8c). Repression of NEDDylation by overexpression of NUB1 also promoted RPL7 interaction with the proteasome, especially under HS conditions (Fig. 8d). In these interaction experiments we did not detect modified RPL7, most likely due to rapid deconjugation during the experimentation, as protease inhibitors were not included in order to isolate active proteasomes. The data suggest that proteotoxic stress promotes modification, at least of a fraction, of RPL7 both with NEDD8 and ubiquitin through HUWE1 and prevents targeting of RPL7 to the proteasome. The data do not exclude the possibility that at least for some RPs, targeting to the proteasome is mediated indirectly, through complex formation with other modified RPs. We then followed an alternative approach to determine the potential role of stress-induced NEDDylation in proteasomal degradation. Under conditions of NEDD8 overexpression, which mimic the stress-induced UBA1-dependent NEDDylation of RPL7 (Supplementary Figure 10A), we found that NEDDylated RPL7 has increased stability compared to ubiquitinated RPL7 (Fig. 9a), consistent with the notion that UBA1-dependent NEDDylation compromises substrate degradation. In addition, we isolated from transfected cells either ubiquitin or hybrid NEDD8-ubiquitin conjugates. These conjugates were then used in vitro to monitor their processing/degradation by purified 26S proteasomes. While ubiquitin conjugates were efficiently processed/degraded, NEDD8/ubiquitin conjugates were almost completely resistant (Fig. 9b). The combination of the above data strongly indicates that NEDD8 and potentially the formation of hybrid NEDD8/ubiquitin chains compromise substrate processing/degradation by the proteasome.

## Discussion

Defining pathways and mechanisms for the formation and elimination of protein aggregates is an important aspect for our understanding of the response to proteotoxic stress. Here, we identify NEDD8 as regulator of the stress-induced nuclear protein aggregation and UPS function. The data suggest that the NEDD8-mediated aggregation protects the nuclear UPS from severe impairment during stress. The finding that both HS and NEDD8 specifically control nuclear but not cytoplasmic UPS activity further supports the notion that NEDD8 is part of the

cellular response to control the function of the nuclear UPS during HS.

The nuclear translocation of damaged substrates for proteasomal degradation has been well characterised in yeast and justified by the high efficiency of the nuclear UPS in these model systems[48–50]. However, it is possible that the majority of NEDDylated substrates upon stress are newly synthesised damaged proteins destined for the nucleus. For example, it is known that RPs upon their synthesis are rapidly imported into the nucleus and the nucleolus[51]. High rates of nuclear import for some substrates may compromise recognition, modification and elimination of such damaged proteins in the cytoplasm.

Another critical aspect is the nature of the observed nuclear aggregates. Seminal studies by Latonen et al. identified similar to the presented nuclear structures upon prolonged inhibition of the proteasome activity. They were reported to contain components of the ubiquitin system, cell cycle regulator proteins and polyadenylated RNA but not rRNA[52]. More recent studies in *Saccharomyces cerevisiae* reported stress-induced peri-nucleolar aggregates, which were previously defined as JUNQ[53]. Our proteomics and immunofluorescence analysis indicate that the observed nuclear structures are nucleolus-related (based on protein composition) but not nucleoli, as we failed to detect key nucleolar markers such as fibrillarin and nucleolin. We speculate that these structures represent a late response to protein damage, involving the reorganisation of the nucleolus.

Our current knowledge on protein NEDDylation defines two modes of NEDD8 activation: The so-called canonical via NAE and the atypical via UBA1 observed under proteotoxic stress conditions[24,25,44]. The use of NAE- and UBA1-specific inhibitors and NUB1 overexpression indicates that the identified role of NEDD8 in stress-induced protein aggregation is independent of the canonical NEDD8 pathway. Rather, it has the main characteristics of the atypical NEDDylation, including the simultaneous modification of targets both with NEDD8 and ubiquitin, possibly by hybrid NEDD8/ubiquitin chains. Based on the findings on RPL7, stress-induced NEDDylation could compete with ubiquitin for substrate modification. This can be either at the level of substrate lysine modification site and/or at the level of ubiquitin chains formation. Indeed, proteomic analysis showed that activation of NEDD8 by UBA1 promotes the NEDDylation of ubiquitin at multiple lysine residues including K48[32]. In this scenario, activation and conjugation of NEDD8 by enzymes of the ubiquitin system could alter the length and/or topology of ubiquitin chains[54], or the so-called ubiquitin code. Stress-induced NEDDylation can thus provide a regulatory mechanism that compromises substrate ubiquitination and targeting for proteasomal degradation (Fig. 10). This is supported by the observation that NEDD8 overexpression, which forces NEDD8 conjugation by the ubiquitin system, reduces the rate of substrate degradation (Fig. 9a) and is sufficient to promote substrate aggregation (Fig. 6a, b). In combination with the in vitro analysis showing a clear preference of the 26S preoteasome for processing/degrading ubiquitin but not hybrid NEDD8/ubiquitin conjugates, the data

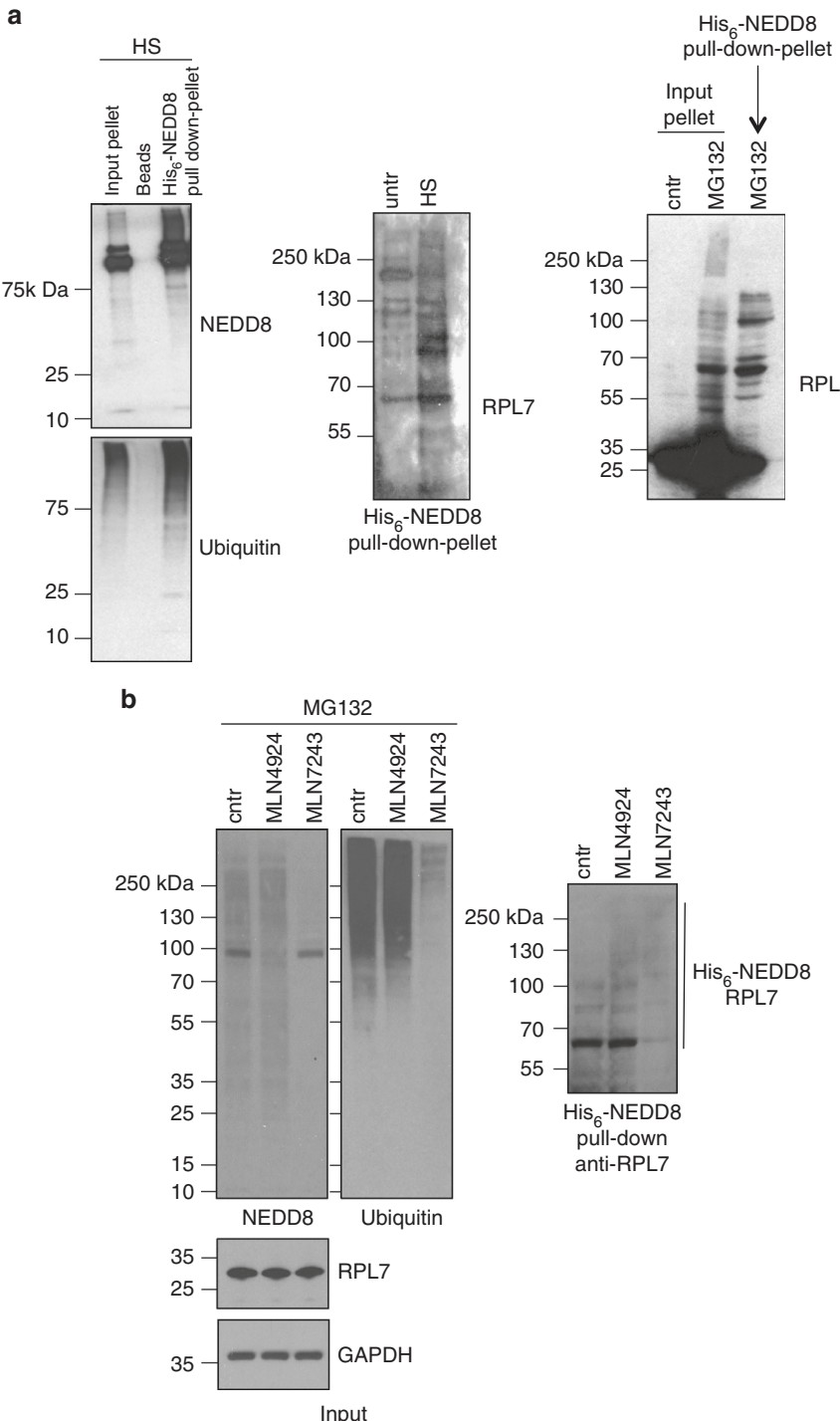

**Fig. 4** UBA1-dependent NEDDylation of RPL7 upon proteotoxic stress. **a** HCT116 cells stably expressing His$_6$-NEDD8 were heat-shocked or treated with MG132 (25 μM, 2 h) and then subjected to a subcellular fractionation. Pellet fractions were solubilized and then NEDDylated substrates were isolated by Ni-NTA pull-down as described in Methods. The isolated NEDDylated substrates were blotted against the indicated proteins. **b** Experiment was performed as above but cells were pretreated with MLN4924 (500 nM 15 h) and MLN7243 (500 nM, 3 h) before MG132 treatment. Ni-NTA pull-down was performed from total cell extracts and western blot analysis as indicated

indicate that the mechanism by which NEDD8 can promote aggregation of misfolded proteins is to prevent their degradation by the proteasome pathway.

The identification of HUWE1 as the key E3-ligase responsible for stress-induced NEDDylation defines a factor of the ubiquitin system that promotes modification of substrates both with NEDD8 and ubiquitin. These findings in combination with the

reported specificity of HUWE1 for RPs upon proteotoxic stress[42] indicate that RPs may be a key target for stress-induced NEDDylation.

While UBA1-dependent NEDDylation is prominent under proteotoxic stress conditions, we cannot exclude that it also operates under homoeostatic conditions, to a decreased extent. It is however evident that UBA1-dependent NEDDylation is

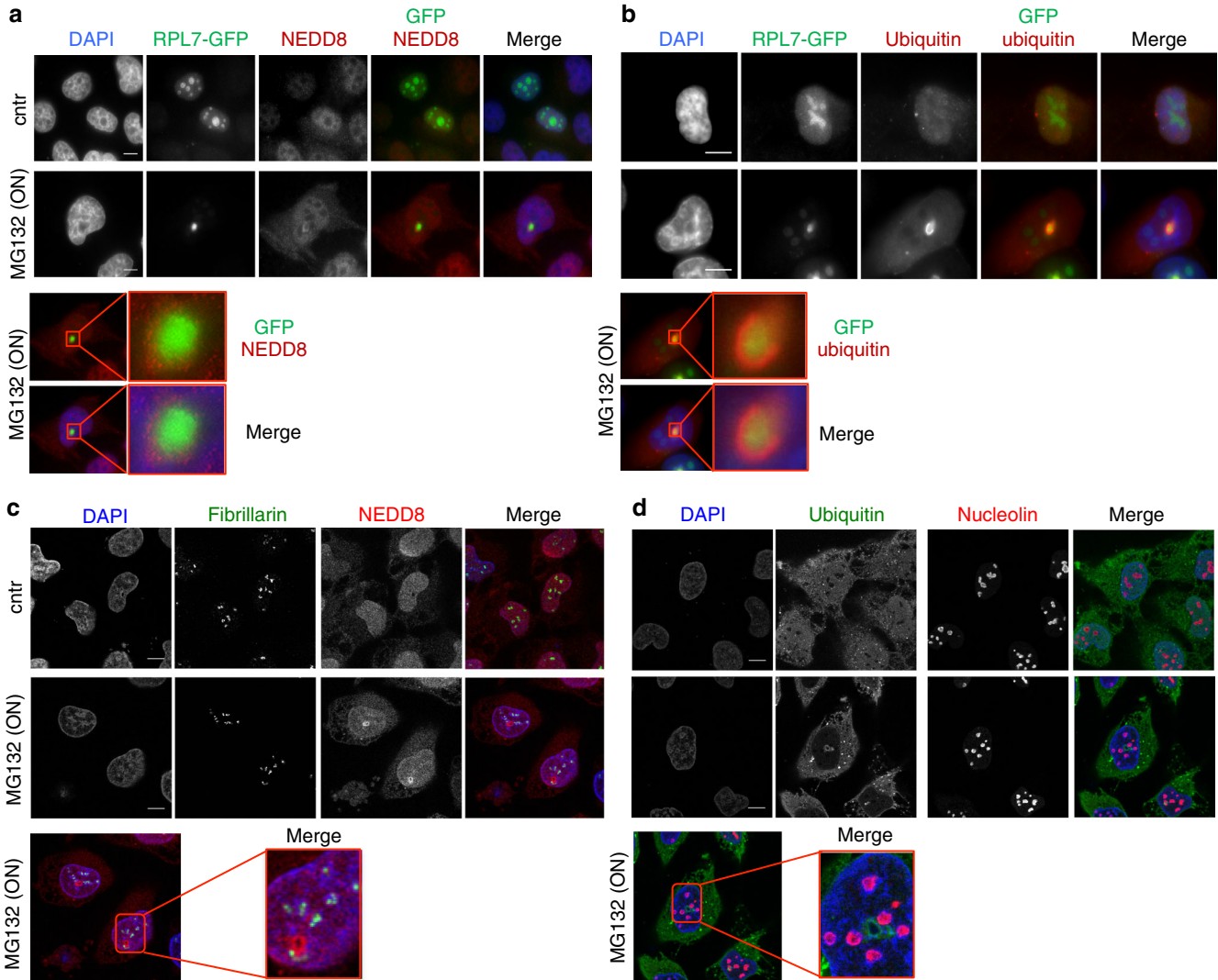

**Fig. 5** RPL7 localises within the NEDD8/ubiquitin-stained aggregates. **a**, **b** H1299 cells were transfected with RPL7-GFP and 48 h later were treated or not with MG132 (5 µM-ON) and stained for either NEDD8 (**a**) or ubiquitin (**b**) (red). Enlarged insets represent the NEDD8/ubiquitin-stained structures observed upon MG132 surrounding RPL7. **c**, **d** Experiment performed as above and cells were stained with fibrillarin and NEDD8 (**c**) or nucleolin and ubiquitin (**d**). Nuclei were stained with DAPI. Scale bars, 10 µm

induced and biologically significant when the burden for proteasomal degradation is dramatically increased during proteotoxic stress. Our studies suggest that the transient nuclear aggregation induced by NEDD8 could indeed act as safeguard mechanism for the UPS during stress conditions.

Consistent with studies in *S. cerevisiae*, and *Caenorhabditis elegans*[46,55,56], our proteomic analysis identifies RPs as a major group of proteins enriched in aggregates upon proteotoxic stress. In addition, we found NEDDylation as a regulatory pathway that promotes their aggregation. A critical aspect of RPs biology is that in their free, non-ribosome state, they are susceptible for rapid proteasomal degradation. RPs are constantly being produced in excess over rRNA and are rapidly degraded by the ubiquitin proteasome pathway in the nucleus[42,46,51]. This provides a significant and constant load for nuclear proteasomes, which however can cope with under homoeostatic conditions. Upon proteotoxic stress where the load of misfolded ubiquitinated proteins targeted for proteasomal degradation is dramatically increased, RPs degradation may now become a limiting factor for proper UPS function. Redirection of a fraction of unassembled/damaged RPs to aggregates via induction of NEDDylation could

provide a means to protect nuclear UPS function during stress (Fig. 10). As discussed above, HUWE1 may be the E3-ligase that provides specificity in this response. It also raises the intriguing possibility that an extraribosomal function of RPs is to impact on UPS function and proteostasis. The study is consistent with the model that the main source of toxicity is the soluble unattended misfolded and possibly oligomeric aggregates and that sequestration into transient insoluble deposits is cytoprotective[20,21,57,58].

## Methods

**Cell culture**. With the exception of H1299 lung carcinoma cells cultured in RPMI medium the rest of cell lines were cultured in Dulbecco's modified Eagle's medium supplemented with 10% fetal bovine serum and standard antibiotics (Penicillin, 50 U/ml and Streptomycin 50 µg/ml) in 5% $CO_2$ at 37 °C in a humidified incubator. HEK293 cells stably expressing NES/NLS-GFPu were a gift from Prof. Ron Kopito and HEK293 RPN11 cells from Dr. Lan Huang. HCT116 and U2OS cells stably expressing wild-type $His_6$-NEDD8 were cultured in the presence of 2.5 µM puromycin for selection. Unless otherwise stated, the American Type Culture Collection was the original source of used cell lines. Cell lines have not been authenticated but were routinely tested for mycoplasma contamination and kept in culture for a maximum of 20 passages.

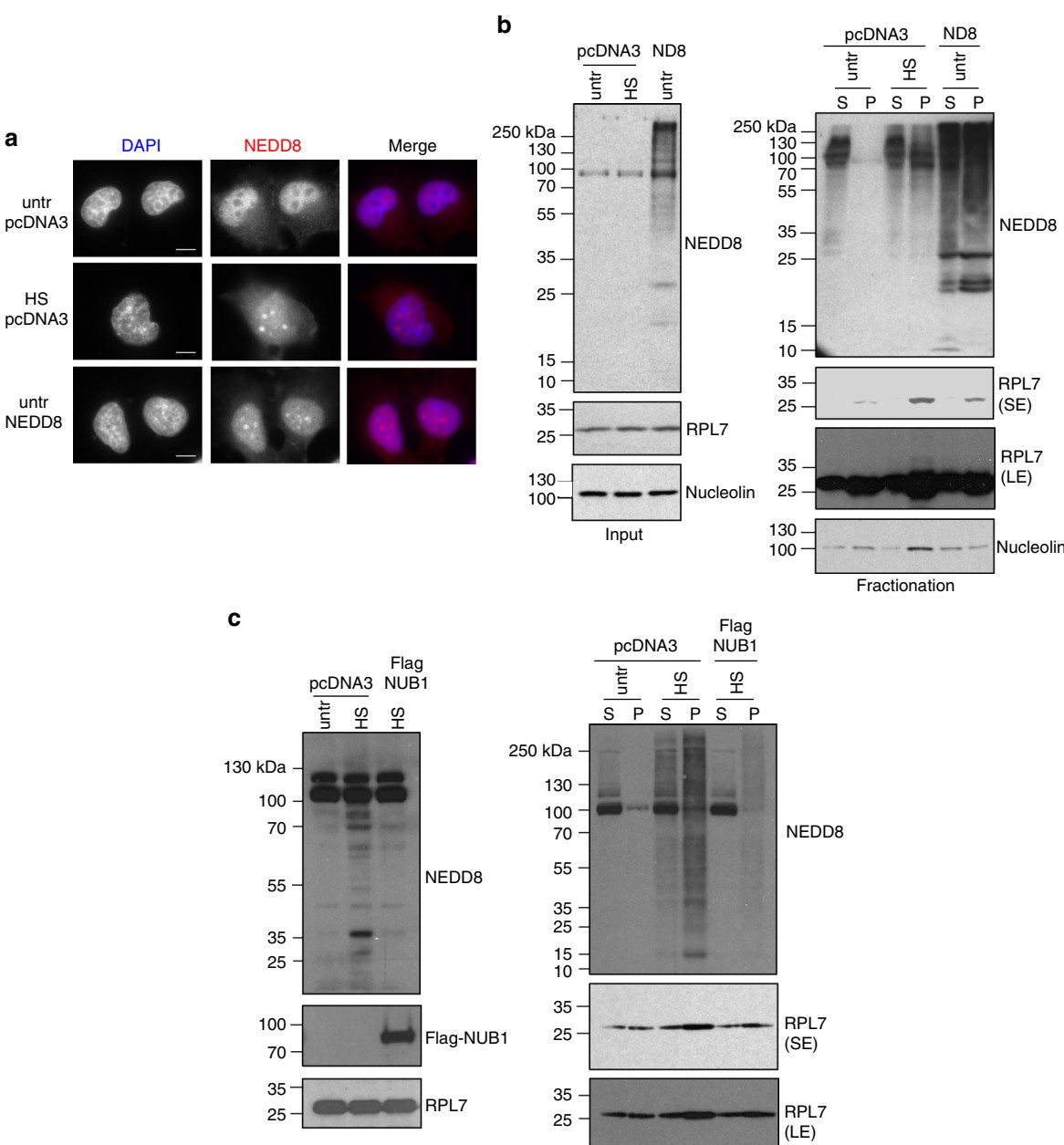

**Fig. 6** Overexpression of NEDD8 promotes protein aggregation. **a** H1299 cells were transfected either with empty pcDNA3 vector or NEDD8 expression constructs. Cells were heat-shocked for 2 h at 43 °C and were stained for NEDD8 and DAPI. Scale bars, 10 μm. **b** Similar experiment as in **a** and fractionation was performed to isolate soluble (S) and insoluble proteins from the pellet (P). Western blotting was performed for the indicated proteins (SE Short exposure, LE Long exposure). **c** Similar experiment as in **a** with the exception that Flag-NUB1 (5μg) construct was transfected

**Reagents**. Most common chemicals were purchased from Sigma Aldrich. MLN4924 (Takeda Pharmaceuticals), MLN7243 (Chemietek), MG132 (Viva Bioscience), Lipofectamine RNAiMAX (Invitrogen), siRNA On-TARGETplus SMARTpools (Dharmacon), protease Inhibitor Cocktail Tablets EDTA-free, Fugene6 HD (Roche), Suc-LLVY-AMC peptide (BostonBiochem). Rabbit mono-clonal anti-NEDD8 (1:2000), Y297 (GeneTex, GTX61205), FK2 mouse anti-ubi-quitin, stainings (1:250) (Viva Bioscience, VB2500), rabbit anti-ubiquitin (1:2000), western blotting (DAKO, Z0458), mouse anti-fibrilarin (1:1000) (ab4566), rabbit anti-nucleolin (1:1000) (ab22758), mouse anti-GAPDH (1:5000) (6C5, ab8245), rabbit anti-RPL7 (1:2000) (ab72550) (Abcam), mouse anti-tubulin (1:2000) (Cell Signalling, 3873), mouse anti-HA (1:2000) (12C5, 11583816001), mouse anti-GFP (1:500) (11814460001) (Roche), mouse anti-a6 proteasome subunit (1 μg/ml) (Enzo Life Sciences, BML-PW8100), rabbit polyclonal anti-HUWE1 (1:2000) (Bethyl laboratories, A300-486A), mouse monoclonal anti-p21 (1 μg/ml) (F-5, sc-6246, Santa Cruz), rabbit polyclonal anti-CDT1 (1:1000) (# 06-1295, Millipore), goat anti-mouse Alexa Fluor® 488 (115-545-146), goat Anti-Rabbit Alexa Fluor® 594 (111-585-008) (Jackson ImmunoResearch).

**Transfections**. Cells were seeded in six-well plates or 10 cm dishes to the desired confluency. 5 nM of siRNA was transfected with Lipofectamine RNAiMAX according to the manufacturer's instructions. Nontarget siRNA was used in control transfections. For the SILAC experiment transfections were performed in six-well plates and 6 h later cells were combined and reseeded in 10 cm dishes. Cells were harvested 48 h post-transfection. Fugene6 HD Transfection Reagent was used for plasmid transfections using 3:1 fugene: DNA (μg) ratio.

**Subcellular fractionation**. After the appropriate treatment, cells in 10 cm culture dish were washed twice with ice-cold PBS, then scrapped into 1 ml PBS. One hundred microliters of cells was pelleted at $16,200 \times g$ for 1 min and lysed with the appropriate volume of 2× SDS loading buffer. The remaining 900 μl of cells was pelleted at 100 x $g$ for 5 min. Cell pellet was resuspended in 300 μl buffer A (10 mM HEPES-KOH pH 8.0, 10 mM KCl, 1.5 mM MgCl$_2$) with protease inhibitor and 10 mM iodoacetamide. Cells were lysed by adding Triton-X100 at a final con-centration of 0.1% for 1 min at 4 °C, then spun for 5 min at $1300 \times g$ 4 °C. The

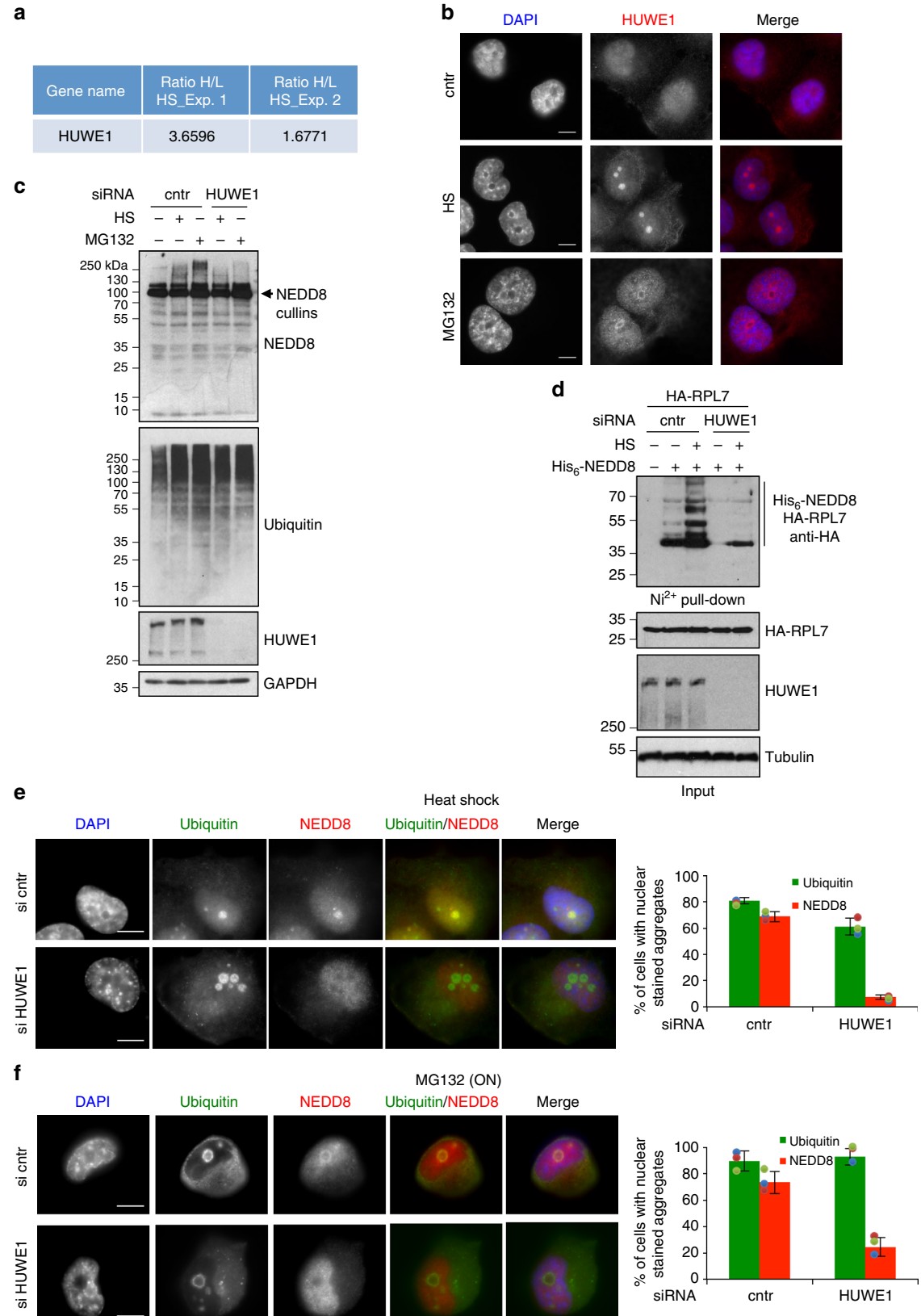

supernatant (cytoplasmic fraction) was mixed with equal volume of 2× SDS. The pellet (nuclear fraction) was washed three times with buffer A, then resuspended with 300 μl of buffer B (20 mM HEPES-KOH pH 8.0, 300 mM NaCl, 2 mM EDTA and 1%NP40) and incubated 30 min on ice. Lysates were sonicated on ice, 8 × 30 s with 30% amplitude (Branson Digital Sonifier) and centrifuged at 16,200 × g, 4 °C

for 15 min. The supernatant (nucleoplasmic fraction) was mixed with equal volume of 2× SDS, and the pellet (pellet fraction) was washed three times with buffer B then resuspended in 2× SDS. For the SILAC proteomics analysis and the experiment in Fig. 6b, c, Supplementary Figure 8 insoluble proteins were directly isolated by lysing cells in 20 mM Tris-HCl pH7.5, 150 mM NaCl, 1.2% deoxycholate, 1.2%

**Fig. 7** The HECT E3-ligase HUWE1 is required for stress-induced NEDDylation. **a** Table showing the SILAC ratio for HUWE1 in the two replicate experiments performed for the aggregate composition upon HS. Heavy, HS; Light, untreated. **b** H1299 cells were either untreated (DMSO) or exposed to HS (2 h) or to overnight MG132 (5 μM) treatment. Cells were fixed and stained for HUWE1. **c** U2OS cells were transfected with siRNAs and treated as above. Cell extracts were used for western blotting. **d** U2OS cells were transfected with siRNAs before plasmid transfection of HA-RPL7 and His$_6$-NEDD8 as indicated. Cells were exposed to HS (2 h) before Ni-NTA pull-down and western blotting. **e, f** H1299 cells were siRNA transfected as indicated and either heat-shocked (**e**) or MG132 (**f**) treated as before. Staining and quantitation of nuclear aggregates stained with NEDD8 or ubiquitin was performed as before. Scale bars, 10 μm. Values represent the average of three independent experiments ± SD

Triton X-100, 200 mM iodoacetamide, complete EDTA-free protease inhibitors. After sonication 5× 30 s at 30% amplitude with 30 s intervals, insoluble proteins were isolated upon centrifugation at 16,200 × g, 4 °C, 30 min. Pellet was washed 3× with lysis buffer before being solubilised in 2× SDS Laemmli's buffer.

**Isolation of NEDD8 substrates from the insoluble fraction.** For all Ni-NTA purification experiments from cell extracts we used approximately 0.8 mg of total protein with the exception of the fractionation experiments where 2 mg of total protein was used as starting material. HCT116 cells stably expressing His$_6$-NEDD8 were subjected to a subcellular fractionation after the appropriate treatment as described above. Pellet fractions were washed three times with the buffer B and then resuspended in 1 ml of 8 M urea, 100 mM Na$_2$HPO$_4$/NaH$_2$PO$_4$ (pH 8), 10 mM Tris-HCl (pH 8) and sonicated 4 × 30 s at 30% amplitude (Branson Digital Sonifier). Insoluble particles were eliminated by centrifugation at 18,800 × g for 10 min at 4 °C. One hundred microliters of each collected supernatant was mixed with equal volume of 2× SDS loading buffer. The rest was diluted up to 6 ml in the 8 M urea pH 8 buffer with 10 mM Imidazole and 10 mM β-mercaptoehanol. The purification was then performed using 60 μl of Ni-NTA agarose beads according to refs. [59,60]. Briefly binding was performed overnight at 4 °C, before 2× washes with 8 M urea, pH 8, 10 mM β-mercaptoehanol, 01% Triton X-100 and 3× washes with 8 M urea, pH 6.3, 10 mM β-mercaptoehanol, 01% Triton X-100. Elution was performed in 2× SDS Laemmli's buffer containing 250 mM imidazole (room temperature, rotating 20 min).

**Immunofluorescence microscopy.** Cells were seeded on round coverslips 24 h before treatment. After the HS/MG132 treatment, cells were washed twice with warm PBS and fixed with 4% formaldehyde for 5 min. Cells were washed three times (10 min) with warm PBS, and permeabilized with 1% Triton X-100 in PBS for 10 min. Cells were washed 3 × 10 min with PBS before blocking with 0.05% Tween-20 + 1% Goat serum in PBS for 1 h. After blockage, cells were incubated with the appropriate primary antibody diluted in 0.05% tween-20 + 1% Goat serum in PBS for 1 h at room temperature (or overnight at 4 °C). After 3 × 10 min washes with 0.05% Tween-20 in PBS, cells were incubated with corresponding secondary antibodies diluted in 0.05% Tween-20 + 1% Goat serum in PBS for 1 h at room temperature in the dark. Samples were washed 3 × 10 min with 0.05% tween-20 in PBS, and then stained with DAPI (1/20,000) for 20 s at room temperature in the dark. Slides were washed 3× with PBS, mounted with Vectashield Mounting Medium (H-1000, Vector), sealed, and viewed under the microscope Leica DM6000 or Leica SP5-SMD using metamorph software. The images were analysed by ImageJ64 software. The 3D rendering experiment was performed from z-stacks (Leica SP5-SMD) using Imaris 8.3.1 (Bitplane an Oxford Instruments company). The colocalisation between NEDD8 and ubiquitin in the nuclear structures was analysed as described in refs. [61,62]. Briefly, quantifications of colocalisation and determination of Pearson's correlation coefficient were performed using the ImageJ JaCoP plugin. For each channel the plot profile was acquired and individual measurements were exported as excel file to make the presented graph. All primary antibodies were used at 1:250 dilution with the exception of anti-NEDD8 (1:150). The secondary antibodies Alexa goat anti-mouse 488 was diluted at 1:500, the goat anti-rabbit 594 1:500.

**Western blot analysis.** For all inputs we routinely used approximately 20 μg of total protein. Proteins were resolved in 4–12% precast Bis-Tris gels and transferred onto PVDF membrane using the Bio-Rad Mini Trans-Blot apparatus. Membranes were blocked in 5% milk solution (PBS with 0.1% Tween-20 and 5% skimmed milk) for 1 h at room temperature with gentle agitation. Membranes were incubated with the primary antibodies overnight at 4 °C. Primary antibodies were diluted in PBS 0.1% Tween-20 with 3% BSA and 0.1% NaN$_3$. Membranes were washed 3 × 10 min with PBS 0.1% Tween-20 and incubated with the appropriate secondary antibody (1:2000) (Sigma Aldrich) for 1 h at room temperature (5% milk). After incubation, membranes were washed 2 × 15 min with PBS 0.1% Tween-20 followed by 2 × 5 min with PBS. Detection was performed with ECL Western Blotting Detection Reagents and membranes were exposed to X-ray Medical Film before being developed. Quantifications were performed with ImageJ. Fully uncropped scans of blots can be found in Supplementary Figure 11.

**SILAC-mass spectrometric analysis.** Cells were labelled either with light (Lys0/Arg0) or heavy (Lys8/Arg10) amino acids. Cells grown in light medium were left at

37 °C whereas cells in heavy medium were heat-shocked at 43 °C for 2 h. In the experiment of NEDD8 knockdown and MLN4924 treatment, control and siNEDD8-transfected cells or control siRNA transfected and MLN4924-treated cells were all heat-shocked at 43 °C for 2 h. For each condition 10 × 10 cm dishes of 80% confluence were collected in PBS counted and equal number of cells were mixed. Upon isolation of insoluble proteins (see subcellular fractionation), 50 μg of protein were run for 15 min on 4–12% precast NuPAGE and coomassie blue stained. Lanes were cut in two gel pieces and in-gel trypsin digestion was performed as described in ref. [63]. Peptides were analysed online by nano-flow HPLC-nanoelectrospray ionisation using a Qexactive Plus mass spectrometer (Thermo Fisher Scientific) coupled to a nano- LC system (U3000-RSLC, Thermo Fisher Scientific). Desalting and preconcentration of samples were performed online on a Pepmap® precolumn (0.3 × 10 mm; Dionex). A gradient consisting of 0–40% B in A for 140 min (A: 0.1% formic acid, 2% acetonitrile in water, and B: 0.1% formic acid in acetonitrile) at 300 nl/min was used to elute peptides from the capillary reverse-phase column (0.075 × 150 mm, Pepmap®, Dionex). Data were acquired using the Xcalibur software (version 4.0). A cycle of one full-scan mass spectrum (375–1500 m/z) at a resolution of 70,000 (at 200 m/z), followed by 12 data-dependent MS/MS spectra (at a resolution of 17,500, isolation window 1.2 m/z) was repeated continuously throughout the nanoLC separation. Raw data analysis was performed using the MaxQuant software (version 1.5.5.1) with standard settings. Used database consists of Human entries from Uniprot (reference proteome UniProt 2017_03) and 250 contaminants (MaxQuant contaminant database). Relative proteins quantifications were calculated on the median SILAC ratios. For further analysis we consider only proteins with at least two peptides. Perseus (version 1.5.5.1) was used for graphical representation of the data, based on the mean value of log2 ratio (only for proteins quantified in duplicate) after elimination of contaminants and reverse entries. Functional enrichment on KEGG pathway was determined using STRING database with a confidence interaction level of 0.7. The False Discovery Rate was calculated from the STRING database analysis on overrepresented KEGG pathways with a confidence interaction level of 0.7.

**$^{35}$S-Methionine labelling and immunoprecipitations.** Cell labelling was performed as described[41]. Briefly, cells were seeded in 6 cm plates and serum starved for 1 h with Methionine/Cysteine-free medium. 350 μCi/plate of $^{35}$S-Methionine was added and cells were either incubated at 37 °C or heat-shocked for 20 min. Cells were washed in PBS, lysed in 400 μl of NP40 lysis buffer and extracts were precleared with protein G beads or GST beads. Immunoprecipitations were performed with 2 μl of NEDD8 antibody or 5 μg of GST-GFPtrap (in house) and 20 μl of either protein G beads or GST beads overnight before washing the beads 3× with 500 μl NP-40 lysis buffer. Samples were eluted in 2× SDS buffer, boiled for 5 min and analysed by SDS-PAGE. Gels were dried and exposed to X-ray films overnight at −70 °C.

**Sequential GST-UBA/ Ni-NTA purification.** 4 × 15 cm dishes of U2OS cells stably expressing His$_6$-NEDD8 were treated with 25 μM of MG132 for 3 h before harvesting in PBS. Cells were lysed in 1 ml of 50 mM Tris-HCl pH 7.4, 150 mM NaCl, 1% NP40, 10% glycerol, 10 mM EDTA, 50 mM iodoacetamide and protease inhibitors (Roche), by syringing 15 times and incubation on ice for 30 min. Upon centrifugation at 16,200 × g for 15 min, supernatant was incubated overnight at 4 °C with 10 μg of GST-UBA (DSK2 protein-in house) and 30 μl of prewashed (lysis buffer) GST beads. Beads were washed 3× with 500 μl lysis buffer (with no protease inhibitors and iodoacetamide) and split in two equal parts. One was used to elute proteins in 2× SDS loading buffer for the UBA input and the other was used to elute proteins (15 min, room temperature) in 500 μl of 8 M urea, 100 mM Na$_2$HPO$_4$/NaH$_2$PO$_4$, 10 mM Tris-HCl pH 8 (Buffer I). The urea eluates were incubated with 50 μl Ni-NTA agarose beads for 4 h at room temperature. Beads were washed 5× with 500 μl of Buffer I and His$_6$-NEDD8 conjugates were eluted with 200 μl of Buffer I, 250 mM imidazole, 15 min at room temperature.

**Quantitative real-time RT-PCR.** Total RNA isolation and SYBR qPCR was performed as described[64] using the following primers:
GFP-F: 5′-ACGTAAACGGCCACAAGTTC
GFP-R: 5′-AAGTCGTGCTGCTTCATGTG,
nedd8-F: 5′- ATGCTAATTAAAGTGAAGAC
nedd8-R: 5′-TCCTCCTCTCAGAGCCAACAC
Briefly, six-well plates were used to isolate total RNA using the SV total RNA isolation kit (Promega). Five hundred nanograms of RNA was used to prepare

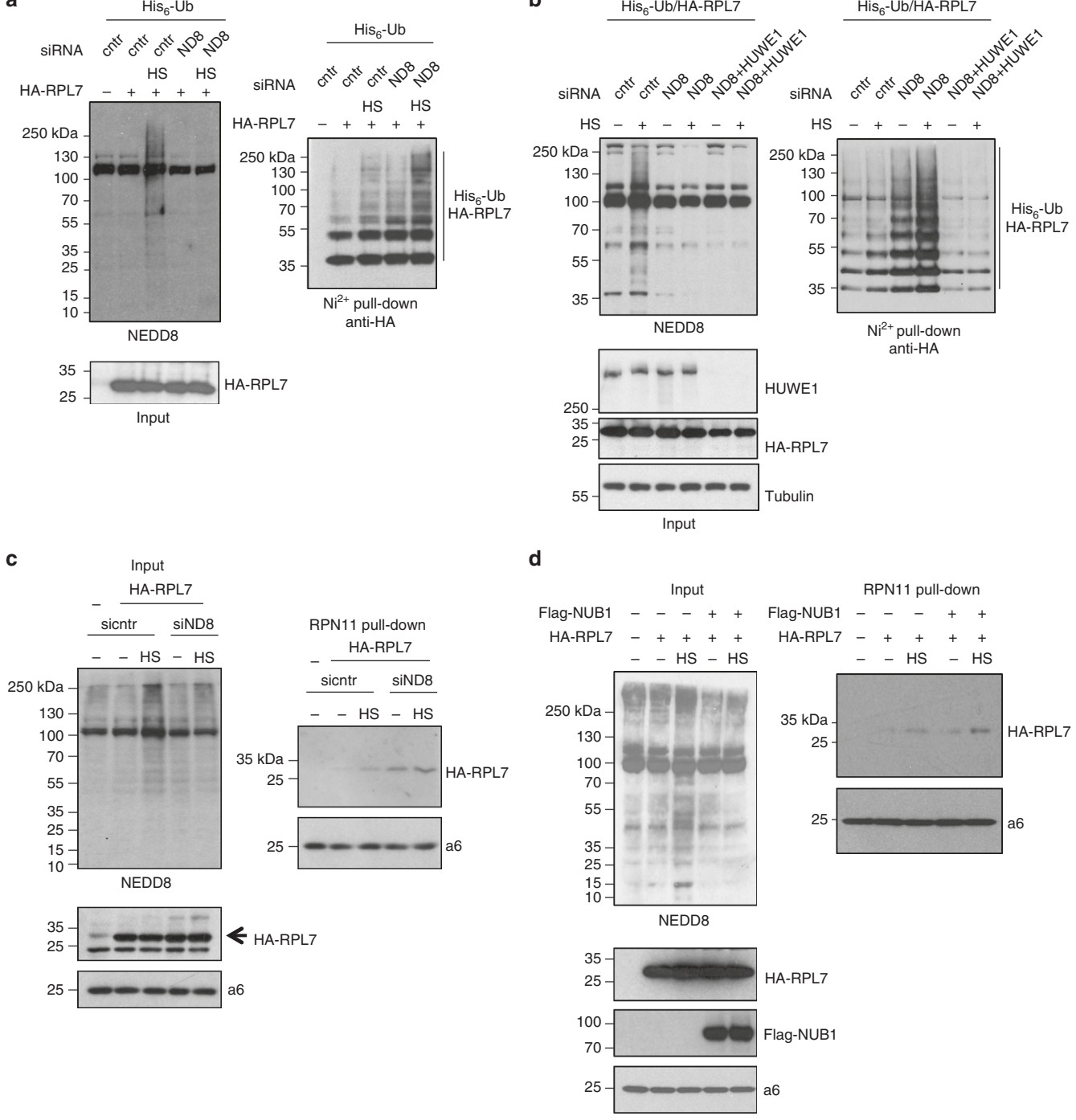

**Fig. 8** NEDDylation compromises substrate ubiquitination and targeting to the proteasome. **a** U2OS cells were first transfected with control or NEDD8 siRNA and 24 h later with His6-ubiquitin and HA-RPL7 plasmids as indicated. Twenty-four hours later cells were heat-shocked for 45 min and extracts were used for Ni-NTA pull-downs. Western blotting on purified His6-ubiquitin conjugates and total cell extracts was performed for the indicated proteins. **b** Similar experiment as in (**a**) but cells were cotransfected with the HUWE1 siRNA as indicated. **c** HEK293 cells stably expressing RPN11-HTBH were transfected with control or NEDD8 siRNA and 24 h later with RPL7 expressing construct. Twenty-four hours later cells were heat-shocked for 45 min and proteasomes were purified with streptavidin beads. Total cell extracts or biotin pull-downs were blotted against the indicated proteins. **d** Similar experiment as in (**c**) with the exception that Flag-NUB1 or pcDNA3 (5 μg) constructs were transfected instead

cDNA with oligo dT using the Transcriptor High Fidelity cDNA Synthesis kit (Roche). 2 μl of 10× diluted cDNA was used in 12 μl qPCR reactions with the appropriate primers and SYBR Green PCR Master Mix (Applied Biosystems).

**Proteasome activity assay in cell extracts**. Proteasome activity in cell extracts was performed according to ref. [65]. U2OS cells in six-well plates were transfected with nontarget or NEDD8 siRNA before reseeded in 10 cm plates. Cells were either untreated or heat-shocked for 30 min and cells were harvested in PBS and lysed in hypotonic buffer containing 10 mM HEPES, pH 7.9, 1.5 mM MgCl$_2$, 10 mM KCl, 0.5 mM DTT, 1 mM ATP, 0.005% digitonin for 2 min on ice. Cells were centrifuged at 1200 × g for 5 min at 4 °C, the supernatant (cytoplasm) was transferred to fresh tubes and pellet was briefly washed with the lysis buffer, passed through a sucrose cushion before lysed in the hypotonic buffer with 0.05% digitonin to obtain the nuclear extract (10 min on ice). Samples were centrifuged at 16,200 × g for 10 min at 4 °C and supernatant (nuclear extracts) were transferred to fresh tubes. 10−20 μg

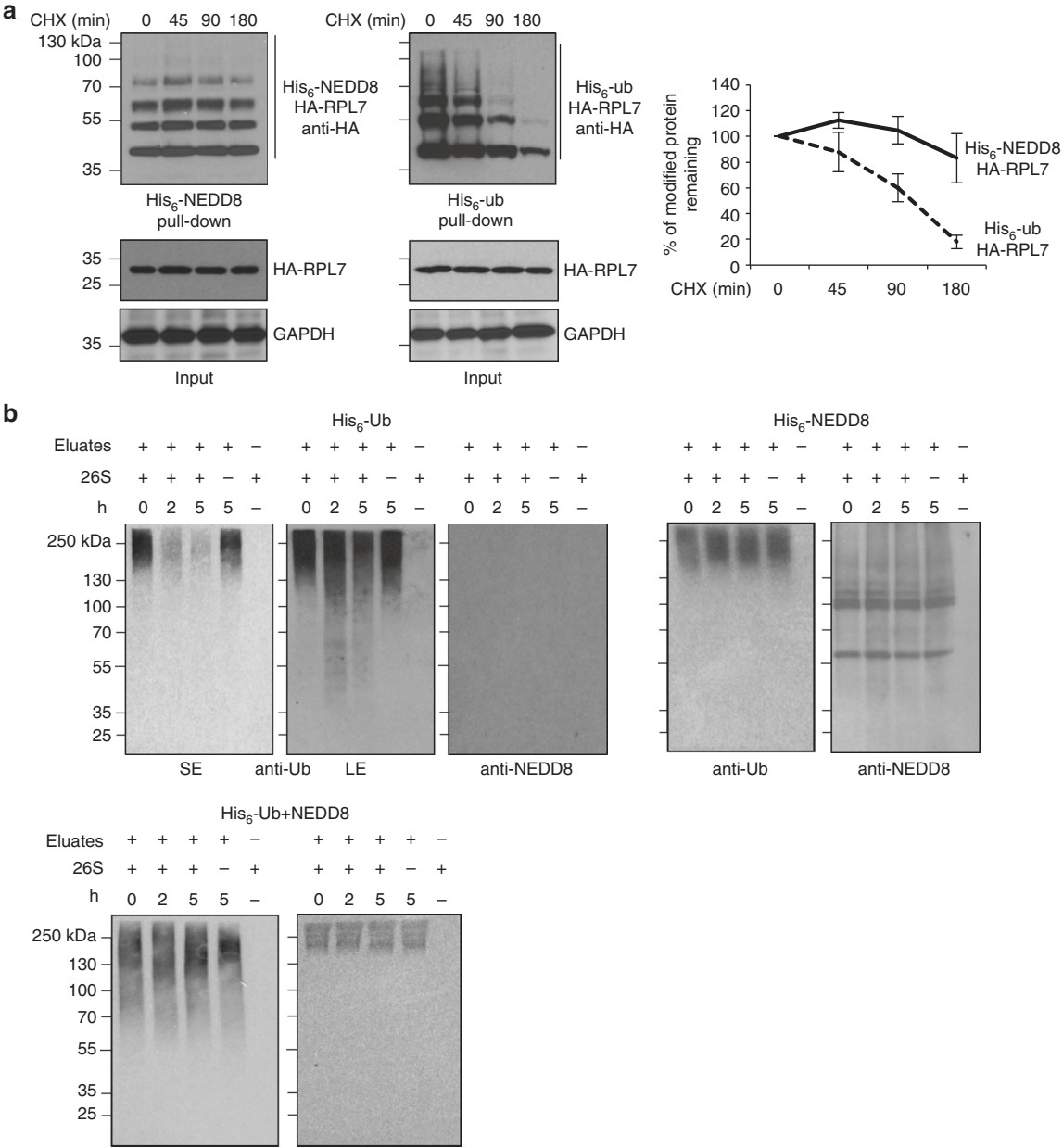

**Fig. 9** NEDDylation compromises proteasomal degradation of substrates. **a** U2OS cells were transfected with 3 μg of HA-RPL7 and 4 μg of His$_6$-NEDD8 or His$_6$-ubiquitin. Forty-eight hours post-transfection cycloheximide (CHX, 100 μg/ml) was added and cells were harvested at the indicated time points, before lysis and Ni-NTA pull-down. The signal in each lane within the length indicated by the bar was used for quantification. The graph represents the average values of three independent experiments ± SD. **b** His$_6$-ubiquitin or His$_6$-NEDD8 conjugates were isolated as described in Methods and used in vitro in a 26S proteasome processing/degradation assay. Western blotting was performed with the indicated antibodies (SE short exposure, LE long exposure)

of total protein was used in a proteasome assay containing 50 mM Tris-HCl pH 7.5, 5 mM MgCl$_2$, 1 mM ATP, 100 μM of the substrate (Suc-LLVY-AMC). Assays were performed in 96-well plates for a period of 30 min. The proteasome inhibitor MG132 was used to measure any nonproteasome protease activity and was subtracted from the data.

**Isolation of conjugates for the in vitro proteasome assays.** 3 × 10 cm dishes of HEK293 cells for each condition were transfected with His$_6$-ubiquitin, His$_6$-NEDD8 or untagged NEDD8 (2 μg each) with the calcium phosphate method as indicated. Forty-eight hours post transfections cells were harvested and lysed in: 25 mM Tris-HCl, pH 7.5, 500 mM NaCl, 0.1% Triton X-100, 1× protease inhibitors (Roche), 50 mM iodoacetamide, incubated on ice for 15 min. After centrifugation at 16,200 × g, 4 °C for 15 min, the supernatant was incubated with 50 μl of Ni-NTA agarose beads, prewashed in lysis buffer, overnight at 4 °C. Beads were extensively washed 5× with 1 ml of 25 mM Tris-HCl, pH 7.5, 1 M NaCl, 0.1% Triton X-100 over a period of 5 h in total at 4 °C, before another 3× washes with the same wash

buffer with no Triton X-100. His$_6$-tagged conjugates were eluted twice with 100 μl (each) of 25 mM Tris-HCl, pH 7.5, 500 mM NaCl, 500 mM imidazole by rotating at 4 °C for 20 min. Eluates were combined and dialysed overnight at 4 °C against 2 l of: 25 mM Tris-HCl, pH 7.5, 10 mM MgCl$_2$. 1 mM of DTT was added to dialysed samples and directly used in vitro in a 26S proteasome assay.

**In vitro proteasome processing/degradation assays.** 26S proteasomes were purified from HEK293 cells stably expressing RPN11-HTBH with streptavidin beads as described in ref. [47]. Briefly, cells from 10 × 15 cm plates were harvested and lysed in lysis buffer: 100 mM NaCl, 50 mM sodium phosphate pH 7.5, 10% glycerol, 5 mM ATP, 1 mM DTT, 5 mM MgCl$_2$, 1× protease inhibitors (Roche) and 0.5% NP-40. Streptavidin beads (100 μl) prewashed in lysis buffer were added to cell extracts and incubated overnight at 4 °C. Beads were washed with 1 ml of the lysis buffer, followed by two washes with 1 ml of wash buffer (50 mM Tris-HCl, pH 7.5, 10% glycerol, 1 mM ATP). Purified proteasomes were released with 1% TEV

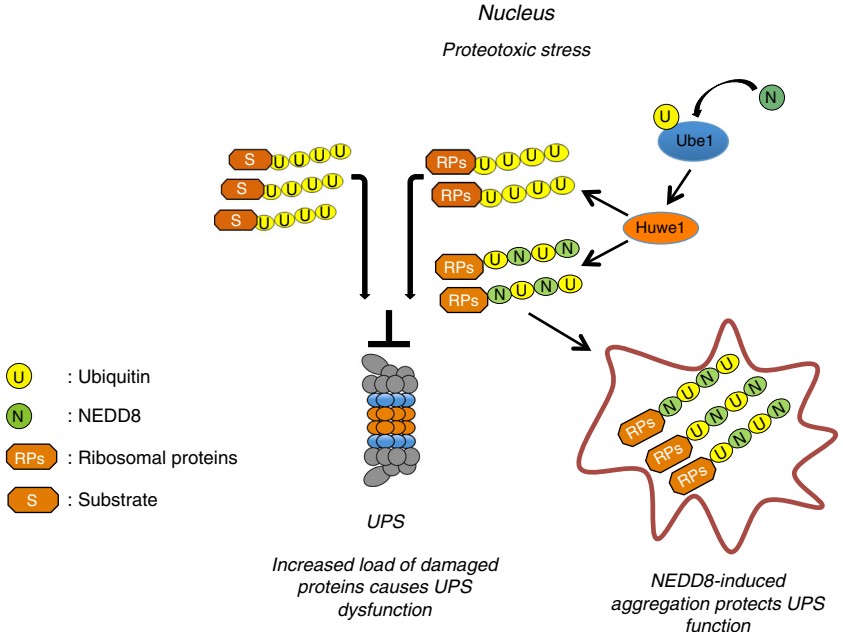

**Fig. 10** Model for the role of NEDDylation in nuclear proteotoxic stress response. Upon proteotoxic stress the load of misfolded ubiquitinated proteins is increased compromising the function of UPS. The activation of NEDD8 by the ubiquitin E1 enzyme UBA1 and its conjugation by E3-ligases such as HUWE1 results in the modification of substrates with NEDD8 and ubiquitin possibly with hybrid NEDD8/ubiquitin chains. This provides a regulatory pathway to prevent substrate targeting to the proteasome, promoting the transient aggregation of a group of proteins. This protects the UPS from the stress-induced toxicity

protease at 30 °C, 1 h. After measuring protein concentration proteasomes were kept at −80 °C.

One microgram of purified 26S proteasomes was used in an in vitro processing/degradation assay with 300 ng of purified His$_6$-tagged conjugates in 40 µl reactions containing 25 mM Tris-HCl, pH 7.5, 10 mM MgCl$_2$, 10% glycerol, 1 mM ATP, 1 mM DTT at 37 °C as described in ref. [66]. Reactions were stopped by the addition of 40 µl of 2× SDS Laemmli's buffer and analysed by western blotting.

### Data availability

The SILAC proteomics data have been deposited in the MassIVE repository (https://massive.ucsd.edu/) with the accession code MSV000082623. All other data supporting the findings of this study are available from the corresponding author on reasonable request.

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

## Acknowledgements

We are grateful to Olivier Coux for his assistance in the proteasome activity assays, the Montpellier Rio Imaging (MRI) facility for the microscopy, Prof. Ron Kopito and Dr. Lan Huang for cell lines. The study was financially supported by the Labex Epi-GenMed, an "Investissements d'avenir" programme, reference ANR-10-LABX-12-01, INCa (PLBIO16-251), the ATIP/AVENIR programme, INSERM and the COST Action (PROTEOSTASIS BM1307-European Cooperation in Science and Technology).

## Author contributions

C.M.M. with the assistance of H.T. performed the majority of the experiments with the exception of the proteomic studies, which were performed by S.L.-G. and the confocal microscopy performed by A.P., S.U. and P.M. performed the proteomic analysis. D.P.X., C.M.M. and S.L.G. designed the experiments with the assistance of M.S.R., D.P.X. supervised the study, wrote the manuscript with the help of all co-authors.

## Additional information

**Competing interests:** The authors declare no competing interests

