## [Peer Review File · Nature Communications]

Reviewer Comments:

Reviewer #1 (Remarks to the Author):

The authors reported that: 1) several cellular stresses induced atypical neddylation of newly synthesized proteins and incorporation of NEDD8 into nuclear aggregates; 2) knockdown of NEDD8 altered the composition of stress-induced nuclear aggregates; 3) atypical neddylation protected the UPS function in the nucleus; 4) ribosome protein RPL7 is a target of atypical neddylation; 5) HUWE1 is a E3 ligase for atypical neddylation. Based on these findings, the authors propose that atypical neddylation protects the nuclear UPS function through promoting nuclear protein aggregation.

Overall, this study addresses an interesting and poorly characterized cellular process. As such, the report is novel. Meanwhile, the work, as presented, appears to have been conducted carefully and thoughtfully. Nevertheless, this reviewer identified a number of issues.

Major issues:

1. This work is mainly conducted in vitro using various cell culture model. One would wonder whether atypical neddylation occurs in vivo and whether the proposed concept is applicable in vivo. In particular, is atypical neddylation observed in any disease models of proteinopathy? would modulations of atypical neddylation have any impact on the UPS function in such models? Answering these questions may improve the significance of this study.
2. It is difficult to understand why silencing NEDD8 would only affect atypical neddylation, which is not supported by the presented data. Therefore, some of the conclusions drawn from siND8 are not very convincing.
3. Per the authors, the NEDD8 response to stress is characterized by (a) the formation of hybrid NEDD8/ubiquitin chains (b) in a Ube1-dependent manner (Line 76-80). However, some of the conclusions were not validated by either means.

Specific:

1. Line 79-80, is it clear that canonical NEDD8 activation via NAE is not affected by any stress condition? Otherwise, it may be more appropriate to state "the atypical" via Ube1 observed under "several" stress conditions,
2. Fig.1B, Puromycin could have effects on multiple cellular processes. To better support the proposed concept that the stress-induced neddylation depends on damage of newly synthesized proteins, additional evidence using genetic means to damage nascent peptides is needed.
3. Fig. 2, the authors claimed that short-term knockdown of NEDD8 fully blocks the stress-induced atypical neddylation but does not affect canonical neddylation (Line 147-151). This is not supported by the blot shown in Fig.2D, in which siND8 caused a reduction of ~100kDa band,

presumably, neddyated cullins (Lane 3 vs Lane 1, Lane 7 vs Lane 5). In fact, a number of neddyated proteins were decreased by siND8 at basal condition (Lane 2 vs Lane 1). Similarly, Fig.3D (Lane 5-8 vs Lane 1-4) and 3E (Lane 7 vs Lane 1) also revealed a decrease in neddyated cullins and other neddyated proteins after knockdown of ND8. Therefore, it is not convincing that silencing NEDD8 can specifically modulate atypical neddylation and thus not appropriate to conclude that atypical neddylation controls the composition of nuclear protein aggregates.

4. Fig.3D, while it is apparent that silencing NEDD8 increased NLS-GFPu, it is not clear that whether silencing NEDD8 and heat shock would affect the transcription and synthesis of NLS-GFPu. Again, as mentioned above, it is not convincing that the increase in NLS-GFPu is specific to the blockade of atypical neddylation.

5. Fig.4A, the authors claimed that RPL7 is the target of atypical neddylation. Would silencing Ube1 also attenuate the neddylation of RPL7 under stress conditions? Moreover, IP of RPL7 under denaturing conditions followed by western blot of both NEDD8 and ubiquitin would tell whether stress does induce NEDD8- and Ub- modification on RPL7.

6. Fig.5, it is quite interesting to propose HUWE1 as an atypical neddylation ligase. The conclusion would be strengthened by testing whether silencing HUWE1 attenuate Ub- and NEDD8- positive species (atypical neddyated proteins) under stress conditions, as done in Fig. 4A.

7. Fig.6, again there is a concern on using siND8 to manipulate atypical neddylation. Would modulations of Ube1 or NUB1L, both known to control atypical neddylation, influence the ubiquitination of RPL7 and the interaction of RPL7 to the proteasome?

Reviewer #2 (Remarks to the Author):

In the manuscript entitled, "Atypical NEDDylation promotes nuclear protein aggregation and protects the Ubiquitin Proteasome System upon proteotoxic stress," Maghames and colleagues present evidence for atypical NEDDylation of ribosomal and nuclear proteins and present arguments supporting that NEDDylation of substrates in a stress dependent manner is a defense mechanism against proteotoxic stress in the cell. They also identify HUWE1 (a E3 ligase) as an integral component for atypical NEDDylation. The results provided in this manuscript claim that concomitant aggregate formation during UPS dysfunction is not only the cause of proteotoxicity but is also a defense mechanism against the same and NEDDylation is the key towards counteracting this stress.

NEDD8 mediated substrate modification is analogous to ubiquitin modification of proteins which is an essential process during protein quality control and turnover. NEDD8 modification of proteins is carried out via both a canonical (NAE, Ubc12, Ube2F) pathway or via an atypical pathway requiring Ube1 instead. Various stressors such as heat shock, proteasome inhibitors as well as oxidative stress have been shown to increase protein NEDDylation however there is less evidence for biological significance of atypical modifications. It has been suggested that NEDDylation is involved in

pathogenesis of various neurological disorders such as Alzheimer's disease essentially via driving the accumulated protein aggregates.

Overall the work is well-done and the experiments are presented well. However, there are major concerns that will need to be addressed to substantiate the central claims in this manuscript.

The main thrust of the manuscript is to characterize the biological significance of what the authors term atypical neddylation. This type of neddylation is distinct from the well-characterized canonical neddylation that uses the Nedd8 heterodimeric E1 enzyme to modify, almost exclusively, cullin substrates in a manner that modulates that activity of all cullin-RING ligases. So-called atypical neddylation occurs when the ubiquitin E1 enzyme, mistakenly charges Nedd8 instead of ubiquitin. This results in ubiquitin E2 enzymes accepting Nedd8 and then utilizing this charged Nedd8 species as they would with ubiquitin to catalyze transfer, in concert with ubiquitin E3 enzymes, to substrates. Essentially, "atypical neddylation" is analogous to ubiquitination except that Nedd8 is used in place of ubiquitin. This has been shown to occur in cells in conditions that lower the concentration of free ubiquitin such that the relative pools of free ubiquitin and Nedd8 now allow for mis-charging of Nedd8 by the ubiquitin E1 enzyme because there is little free ubiquitin to charge. The ability of the ubiquitin E1 to utilize Nedd8 has been well-established. Here the authors use heat-shock which, like proteasome inhibition, results in a lowering of free ubiquitin levels that allow for an increase in atypical neddylation. What the biological significance, if any, of this atypical neddylation is not well established.

Major issues:

1) The tools to differentiate between canonical neddylation and atypical neddylation in this study are utilization of the Nedd8 E1 inhibitor MLN4924 to inhibit canonical neddylation and siRNA-mediated knockdown of Nedd8 to inhibit atypical neddylation. While it is extremely clear that MLN4924 treatment inhibits canonical neddylation it is less clear that knockdown of Nedd8 only impacts atypical neddylation. In fact, no evidence is presented that knockdown of Nedd8, which one would predict would impact ALL neddylation, only inhibits atypical neddylation. The authors make an argument that knockdown of nedd8 only inhibits atypical neddylation in data presented in figure 2D. They argue that knockdown of nedd8 reduces the amount of high molecular weight neddylated species that are induced upon heat shock (which is certainly true) but does not reduce canonical neddylation. The second claim is not substantiated by the data as there is a clear reduction in the amount of cullin neddylation (dark band just above 100kD) upon knockdown of Nedd8 in both untreated and heat-shocked conditions. As such, knockdown of Nedd8 is effecting BOTH canonical neddylation and atypical neddylation making any claims about the function of atypical neddylation using knockdown of Nedd8 impossible to interpret. How do we know that any of the effects on aggregation or turnover of reporter proteins (NLSGFPu) arise from inhibition of canonical neddylation. There are certainly hints that there are real differences here. This is best represented by figures 2F-H where knockdown of nedd8 appears to increase protein "aggregation" whereas MLN4924 treatment has a much different effect. There appears to be some evidence that mixed Nedd8-Ub chains may alter the solubility of proteins, but MUCH more biochemical evidence is needed to support this claim, especially due to the ambiguity of the Nedd8 knockdown approach.

2) The authors routinely utilize a biochemical separate technique to differentiate soluble from aggregated protein. While this is a good idea, the technique that the authors use comes

“aggregated” proteins with chromatin. It is clear that there is chromatin in the pellet fraction (Figure 1C) as well as ubiquitinated and neddylated protein. As such, this fraction likely contains other proteins that are not aggregated so any claims that proteins identified in this fraction by mass spectrometry are aggregated or less-soluble than are not substantiated by the data.

3) The significance of the Ring-like nuclear structures are not well-established. Certainly, nedd8 and ubiquitin localize to these structures that seem to encircle RPL7-GFP, but it is not clear if this is some sort of nuclear quality control compartment. Are other ribosomal proteins present within the Ring? Are other cytoplasmic proteins found to be de-enriched in pellet fractions upon Nedd8 knockdown found in those rings? Are other well-characterized nuclear aggregating proteins found in these rings? Is the formation of these rings upon heat-shock blocked by inhibition of the ubiquitin E1 or the Nedd8 E1?

4) The SILAC experiments are very hard to interpret. The authors enrich for pellet fraction proteins upon heat shock and find nearly every highly abundant protein has a \log_2 ratio >1 . This result is not surprisingly in the least, as heat shock will drive protein aggregation and nearly every protein will be found in this pellet fraction. In fact, their distribution shown in Figure 2B is entirely skewed to the right. The authors chose to highlight ribosomal proteins in this plot for unknown reasons as almost every highly abundant protein can be found in this list, Tubulin, proteasome components, metabolic enzymes, etc. The choice to highlight ribosomal proteins is not well-established and seems random as there many proteins whose \log_2 ratio is >1 in this experiment. The exact same statement can be made about the data presented in figure 2F. Almost every protein identified shows less presence in pellet fractions upon nedd8 knockdown. This result is very non-specific (again as the data is entirely skewed), so why focus on nuclear proteins or ribosomal proteins? Do ribosomal proteins even pass a significance test here given that the entire population is skewed to the left? They seem to be in the bulk of the population? What proteins are 2 or 3 standard deviations away from this skewed mean?

5) The authors utilize cytoplasmic or nuclear localized version of unstable GFP (GFPu) to attempt to show that atypical neddylation is required for the efficient turnover of nuclear ubiquitin-proteasome substrates. They clearly show that NLSGFPu but not NESGFPu accumulates upon heat shock. The authors don't comment on the reason for this difference. Both substrates would utilize the exact same enzymes to catalyze their turnover as they have the same degron, so why would they behave differently upon heat shock? No explanation is given. Further, the authors claim that the accumulation of NLSGFPu upon heat shock is exacerbated upon Nedd8 knockdown and this is entirely due to some mechanism that relies on atypical neddylation. However, this is based on small differences between Nedd8 knockdown and control knockdown (24.1 fold vs 29.8). These kinds of differences can arise from differences in western blotting which has limited quantitative resolution (especially if the authors are using film for their immunoblots). I would want to see true biological replicate experiments showing a reliable and truly quantitative difference between NLSGFPu levels upon heat shock for this result to be meaningful.

6) The authors seem to be making an argument that Neddylation drives proteins toward aggregation and using experiments like those depicted in figure 4 to make that argument. While it is clear that there is more neddylated RPL7 in the pellet upon heat shock, is this specific for nedd8 or RPL7? What if the authors repeated this experiment using His-Ubiquitin. Would the result be different? What about for any of the hundreds of other proteins whose presence in the pellet fraction increases upon

heat shock (like tubulin, or GAPDH?). If this is just mimicking ubiquitination, then atypical neddylation is merely a biochemical artifact.

7) The idea that atypical neddylation drives proteins to aggregate and thus protect the ubiquitin proteasome system is complicated by the authors own data. In figure 4G, the authors overexpress Nedd8, which, according to the authors hypothesis, should drive proteins toward aggregation. However, there is clearly less RPL7 in the pellet fraction upon Nedd8 overexpression. This argues that atypical neddylation prevents aggregation of at least RPL7.

8) All the data with Huwe1 merely represents a biochemical phenomenon that takes place upon lowering free ubiquitin levels (like upon heat shock). Because the ubiquitin E1 is mistakenly utilizing Nedd8 instead of ubiquitin, many, if not all, ubiquitin E2 enzymes will accept this activated Nedd8 from the Ubiquitin E1 enzyme and then utilize this Nedd8 as it would ubiquitin in all transfer reactions with E3 enzymes, like Huwe1. Thus, ANY ubiquitin ligase would show increased transfer of Nedd8 to its substrates under these conditions, including Huwe1. The authors are just merely demonstrated a biochemical aberration that occurs upon heat shock that could be demonstrate for any ubiquitin ligase/substrate pair.

9) In order for the authors to argue that atypical neddylation alters the solubility and turnover of proteins in which atypical neddylation occurs, the authors need to actually demonstrate either of those outcomes. For instance, does the rate of turnover, using metabolic pulse chase assays, of RPL7 or some collection of putative atypical neddylation substrates, change, at steady-state, and upon heat shock upon nedd8 knockdown. Even this result would be a bit inconclusive due to the inherent problems in nedd8 knockdown (see point 1). Further, does a protein with a mixed nedd8-ubiquitin chain get destroyed by the proteasome at a different rate than the same protein with a pure ubiquitin chain (of the same length). This kind of biochemical demonstration of a difference between a mixed nedd8-ub chain and a pure ub-chain would be required to begin to clearly demonstrate a role for these mixed-chains (which clearly can form in cells) in regulating protein turnover.

Minor points:

The immunofluorescence microscopy in Figure 1E with DAPI staining is hardly visible and most of the microscopy should show a gray scaled version of each panels. A 3D surface rendering would be a good way of showing effective co-localization. siRNA treatment of NEDD8 coupled with immunofluorescence microscopy might also be a good option.

Demonstrating NEDD8 co-localization with RPL7-GFP. Showing similar data with another ribosomal protein such as RPL8 or RPL11 should strengthen the argument.

Page 5 – Typo – “This study” instead of “The studies”

Figure 4B, C, E – Inset panel points to MG132 treatment and needs to be labeled.

Pearson’s Coefficient calculate in Figure 1 is not described in the methods section.

How was the FDR calculated in Figure S2?

How was the relative GFP level calculated in Figure 3? ImageJ? Was film used?

Reviewer #3 (Remarks to the Author):

This manuscript reports the atypical neddylation of newly synthesized proteins that are either misfolded or form aggregates during the heat shock treatment. Using metabolic labeling (SILAC), the authors observed an enrichment in NEDD8 in the insoluble pellet of cells that were treated to a heat shock, suggesting a role for protein neddylation in protein aggregation. Out of the ~1700 proteins quantified, they identified a subset of ~ 55 ribosomal proteins in the insoluble fraction upon heat shock treatment. By analyzing the turnover of NES- and NLS-GFP constructs they determined that the nuclear proteasome is impaired during the heat shock response, an effect that is accentuated by knocking down the NEDD8 machinery. Follow-up experiments on RPL7 confirmed the neddylation of this substrate during heat stress and its increased aggregation when Nedd8 is overexpressed. Immunofluorescence microscopy experiments confirmed that HECT E3 ligase HUWE1 colocalized in heat shock-induced aggregates, and knock down of this ligase also reduced neddylation of RPL7, thus suggesting a role for HUWE1 in atypical protein neddylation. By knocking down NEDD8, the authors found an increased ubiquitination of RPL7 indicating that neddylation competes with ubiquitination during heat shock. Altogether, these results suggest that atypical NEDD8 conjugation may protect substrates from proteasome degradation during heat shock by favoring protein aggregation.

Overall, the manuscript provides valuable information on the potential interplay between neddylation and ubiquitination during heat stress. The authors provide appropriate data to support their claims, though several experiments lack replicates to evaluate the statistical significance of their findings. For example, all SILAC experiments are conducted on single injection with no replicates, and more than 10% of abundance measurements are obtained for protein quantified with only one peptide. Other reproducibility measurements should be provided for immunofluorescence microscopy experiments. Also, there is an overwhelming number of figure panels (Figures 1 and 4), and the authors should make an effort to move non-essential display items to supplementary material. Additional points are outlined below:

1. Figure 1, panel H) should be moved to the supplemental. For Panel E) there should be a bar graph to show that these ring formations are statistically significant in the MG132 treated cells. Error bars should be provided for panel G).

2. In Figure 2B and the accompanying text, it is surprising that the authors do not comment on the interplay with other UBLs such as SUMO. Interestingly, their supplementary table reports the

occurrence of all three SUMO paralogs in aggregates, raising the possibility that protein sumoylation may also contribute as previously reported for heat shock treatment (e.g. *Sci Signal.* 2009 May 26;2(72):ra24; *Cell Div.* 2015 Jun 20;10:4). A comment regarding the significance of protein modification by other UBLs is warranted.

3. Replicate SILAC experiments should be reported for data shown in Figures 2 B), F), G), and H) to determine the statistical significance of abundance changes measured.

4. The amount of cell extracts used for immunoblots in the input and after NTA purification should be reported.

5. For Figure 4, why is ubiquitin found in the nucleus of MG132-treated cells in panel C) and in the cytoplasm in panel E)?

6. For Figure 4 G), in the right bottom panel, why is the pelleted RPL7 not conjugated by Ubi or NEDD8? The mass shown on the blot corresponds to the unmodified form. Why is the signal so faint compared to the input?

7. For Figure 5 C), is there a less exposed version of the blot that would allow to quantitate the conjugated cullin?

8. Figure 6 C), in the right panel, why is the RPL7 not modified by NEDD8 or Ubiquitin (no mass shift by gel). Could it be that RPL7 is not actually neddylated, and that but another protein from the ribosomal complex is?

9. For Figure 6 B), right panel, the membrane was blotted with what antibody? Is it HIS or HA?

10. Check text for inconsistency and typos (Line 482 "Humanentries" should be "Human entries")

We would like to thank all reviewers for their time in reviewing our manuscript. We believe that based on their comments we have significantly improved the manuscript. Detailed response to comments:

Reviewer #1 (Remarks to the Author):

The authors reported that: 1) several cellular stresses induced atypical neddylation of newly synthesized proteins and incorporation of NEDD8 into nuclear aggregates; 2) knockdown of NEDD8 altered the composition of stress-induced nuclear aggregates; 3) atypical neddylation protected the UPS function in the nucleus; 4) ribosome protein RPL7 is a target of atypical neddylation; 5) HUWE1 is a E3 ligase for atypical neddylation. Based on these findings, the authors propose that atypical neddylation protects the nuclear UPS function through promoting nuclear protein aggregation.

Overall, this study addresses an interesting and poorly characterized cellular process. As such, the report is novel. Meanwhile, the work, as presented, appears to have been conducted carefully and thoughtfully. Nevertheless, this reviewer identified a number of issues.

Major issues:

1. This work is mainly conducted in vitro using various cell culture model. One would wonder whether atypical neddylation occurs in vivo and whether the proposed concept is applicable in vivo. In particular, is atypical neddylation observed in any disease models of proteinopathy? would modulations of atypical neddylation have any impact on the UPS function in such models? Answering these questions may improve the significance of this study.

Response

The accumulation of NEDD8 in aggregates in neurodegenerative diseases was initially reported almost 15 years ago. However, these *in vivo* observations have remained unexplored since then. More recently, NUB1, which interacts with NEDD8 and ubiquitin chains, was identified as a regulator of tau aggregation in Alzheimer's disease (Richet et al., 2012) and Huntingtin toxicity (Lu et al., 2013) (please also see below regarding our studies on NUB1).

We fully agree that characterization of stress-induced NEDDylation *in vivo* is important. However, new animal model systems have to be generated including hypomorphic mutations in UBA1 and potentially testing the UBA1 and NAE inhibitors in proteinopathy animal models. Protocols for the use of such inhibitors in proteinopathy animal models do not currently exist and have to be established. Our laboratory has no expertise in the use of animal model systems and is something we will have to set up in a long-term collaboration, which we feel goes beyond the scope and time frame of this study.

In the introduction, we describe the “forgotten” and more recent studies on NEDD8 and neurodegenerative diseases and we hope that our findings on the biology and mechanisms of formation of NEDD8 protein aggregates will re-ignite research in this area.

2. It is difficult to understand why silencing NEDD8 would only affect atypical neddylation, which is not supported by the presented data. Therefore, some of the conclusions drawn from siND8 are not very convincing.

3. Per the authors, the NEDD8 response to stress is characterized by (a) the formation of hybrid NEDD8/ubiquitin chains (b) in a Ube1-dependent manner (Line 76-80). However, some of the conclusions were not validated by either means.

Specific:

1. Line 79-80, is it clear that canonical NEDD8 activation via NAE is not affected by any stress condition? Otherwise, it may be more appropriate to state “the atypical” via Ube1 observed under “several” stress conditions,

Response

We agree with the reviewer’s comment. Our observations on stress conditions are on proteotoxic stress, which do not appear to affect cullin-neddylated and CRL function. We have now re-phrased similar sentences in the text.

2. Fig.1B, Puromycin could have effects on multiple cellular processes. To better support the proposed concept that the stress-induced neddylated depends on damage of newly synthesized proteins, additional evidence using genetic means to damage nascent peptides is needed.

Response

We now include an experiment with we use the specific HSP70 inhibitor VER-155008. Short-term inhibition (5-6hrs) of HSP70 induces the production of damaged newly synthesized proteins. We found that similarly to heat shock, but to a decreased extent, HSP70 inhibition induces NEDDylation. We feel that the combination of the two approaches suggest that the observed induced NEDDylation upon heat shock is, at least in part, due to modification of damaged newly synthesized proteins. This conclusion is also supported by the kinetics of the response presented in the fractionation experiment in Fig. 1C. Here, we observe a progressive accumulation of NEDDylated proteins from the cytoplasm into the nucleus.

3. Fig. 2, the authors claimed that short-term knockdown of NEDD8 fully blocks the stress-induced atypical neddylated but does not affect canonical neddylated (Line 147-151). This is not supported by the blot shown in Fig.2D, in which siND8 caused a reduction of ~100kDa band, presumably, neddylated cullins (Lane 3 vs Lane 1, Lane 7 vs Lane 5). In fact, a number of neddylated proteins were decreased by siND8 at basal condition (Lane 2 vs Lane 1). Similarly, Fig.3D (Lane 5-8 vs Lane 1-4) and 3E (Lane 7 vs Lane 1) also revealed a decrease in neddylated cullins and other neddylated proteins after knockdown of ND8. Therefore, it is not convincing that silencing NEDD8 can specifically modulate atypical neddylated and thus not appropriate to conclude that atypical neddylated controls the composition of nuclear protein aggregates.

Response

We believe that the most appropriate approach to specifically block stress-induced NEDDylation without affecting global ubiquitination is the NEDD8 knockdown. We fully agree that NEDD8 knockdown has an effect on NAE dependent NEDDylation. However, by using cullin-NEDDylation as marker for NAE dependent NEDDylation we believe the effect is relatively minor compared to the effect on the stress-induced NEDDylation. While NEDD8 knockdown causes a small decrease in cullin NEDDylation, it does not impact on the CRL E3-ligase activity, which is the relevant biological outcome. This is supported by previous studies using the same approach

(Sundqvist et al., 2009) and by data provided in S2A: Here, we found that NEDD8 knockdown has no effect on the levels of well-established substrates for CRLs (p21, CDT1) indicating that CRLs are not affected. These results are also consistent with studies indicating that a modest decrease in cullin NEDDylation has no effect on CRL activity (Scott et al., 2017). Indeed, almost a complete inhibition of cullin NEDDylation is required to effectively block CRL function (Soucy et al., 2009). This can be achieved with the NAE inhibitor MLN4924, as shown in Fig. 2D and S2A.

While these observations suggest that NEDD8 knockdown mainly impacts on stress-induced NEDDylation, defects due to inhibition of NAE dependent NEDDylation cannot be excluded. To discriminate these effects we compared the siNEDD8 data with the effect of MLN4924 treatment, which blocks NAE-dependent but not stress-induced NEDDylation. This comparison should reveal the effect of NEDD8 knockdown on stress-induced NEDDylation. We believe the used approach is currently the most appropriate to specifically determine and discriminate the effects of NEDDylation in homeostatic (NAE dependent) and proteotoxic stress conditions (UBA1 dependent). In addition, the new immunofluorescence data on the effect of NAE and UBA1 inhibitors and HUWE1 knockdown strongly support the notion that the observed effects are due to the so-called “atypical” NEDDylation.

However, we do acknowledge that the readers of the manuscript may have similar concerns. To clarify this issue, we have now added in the manuscript the above arguments/clarifications and decided to tone-down the manuscript: We now refer to our observations as the effect of NEDDylation in proteotoxic stress response. In the discussion section, based on the presented evidence, we describe the arguments that the process we monitor has all the characteristics of “atypical” NEDDylation and provide a hypothesis on the biological role of “atypical” NEDDylation in the proteotoxic stress response.

4. Fig.3D, while it is apparent that silencing NEDD8 increased NLS-GFPu, it is not clear that whether silencing NEDD8 and heat shock would affect the transcription and synthesis of NLS-GFPu. Again, as mentioned above, it is not convincing that the increase in NLS-GFPu is specific to the blockade of atypical neddylation.

Response

We monitored the effect of NEDD8 knockdown and MLN4924 treatment both on mRNA levels and protein synthesis of NLS-GFPu (S3). We found that the observed effects on NLS-GFPu levels are not due to transcriptional or protein synthesis changes. We would like to note that the levels of a very similar construct (NES-GFPu) are not affected by siNEDD8. In addition, the observed effects on the NLS-GFPu levels are specific to siNEDD8 as MLN4924 treatment has no effect.

5. Fig.4A, the authors claimed that RPL7 is the target of atypical neddylation. Would silencing Ube1 also attenuate the neddylation of RPL7 under stress conditions?

Response

With the use of specific NAE and UBA1 (UBE1) inhibitors, we now show in Fig. 4B that RPL7 NEDDylation depends on UBA1 but not on NAE, a key characteristic of “atypical”

NEDDylation.

Moreover, IP of RPL7 under denaturing conditions followed by western blot of both NEDD8 and ubiquitin would tell whether stress does induce NEDD8- and Ub-modification on RPL7.

Response

In Fig. 4, 5D, 6 we had shown that RPL7 is modified with NEDD8 and ubiquitin. However, the used approaches do not demonstrate the simultaneous modification of the substrate with NEDD8 and ubiquitin. We feel this issue cannot be addressed by the proposed IP approach, as distinct pools of RPL7 may exist, modified either with NEDD8 or ubiquitin.

We therefore used an alternative method based on previous studies, showing that UBA domains apart from isolating poly-ubiquitinated substrates can be also used to isolate hybrid NEDD8/ubiquitin conjugates induced upon proteotoxic stress (Leidecker et al., 2012). We used extracts from His₆-NEDD8 stable cells exposed to proteotoxic stress to isolate hybrid NEDD8/ubiquitin conjugates with the UBA domain from DSK2. Eluates were then used for a second purification of His₆-NEDD8 conjugates under denaturing conditions. We observe the modification of RPL7 in both eluates, providing strong evidence for the simultaneous modification of the substrate with NEDD8 and ubiquitin, possibly with hybrid chains.

The observation that we detect mainly high-molecular weight RPL7 conjugates is most likely due to the high affinity of UBA domains for poly-modified conjugates, which are preferentially enriched over mono-modified conjugates.

6. Fig.5, it is quite interesting to propose HUWE1 as an atypical neddylation ligase. The conclusion would be strengthened by testing whether silencing HUWE1 attenuate Ub- and NEDD8- positive species (atypical neddylated proteins) under stress conditions, as done in Fig. 4A.

Response

Indeed, we found this experiment particularly informative. Consistent with the western blot analysis, knockdown of HUWE1 causes a dramatic decrease in NEDD8 stained aggregates induced by proteotoxic stress (Fig. 5E, F). Importantly, the effect is rather specific as the cytoplasmic and nucleoplasmic NEDD8 staining is not affected (very similar to siNEDD8 effect). In addition, no dramatic effect is observed on ubiquitin stained nuclear aggregates. This is most likely due to the redundancy of the response, as additional E3-ligases promote only ubiquitination of substrates during proteotoxic stress, in the absence of HUWE1. In addition, HUWE1 has substrate specificity, as RPs were recently reported as the main substrate during proteotoxic stress. Thus, the dramatic effect of HUWE1 knockdown on stress-induced NEDDylation indicates the specificity of the response, with HUWE1 being the key if not the only E3 that promotes “atypical” NEDDylation. To our knowledge HUWE1 is the first identified E3 that controls stress-induced NEDDylation.

Upon heat shock we observe approximately 20% decrease in ubiquitin stained aggregates upon HUWE1 knockdown, which may represent the hybrid NEDD8/ubiquitin conjugates. As the nuclear stained aggregates are scored in a qualitative and not

quantitative manner, this decrease (20%) of potentially hybrid NEDD8/ubiquitin aggregates could be an underestimation.

7. *Fig.6, again there is a concern on using siND8 to manipulate atypical neddylation. Would modulations of Ube1 or NUB1L, both known to control atypical neddylation, influence the ubiquitination of RPL7 and the interaction of RPL7 to the proteasome?*

Response

Indeed, the role of NUB1L/NUB1 in protein aggregation is rather interesting, as studies indicate a role in Alzheimer's disease (Richet et al., 2012) and Huntington toxicity (Lu et al., 2013). However, the key aim in this experiment is to specifically determine the role of NEDD8 in RPL7 ubiquitination and proteasome targeting. Modulation of UBA1, NUB1L or NUB1 does not only affect protein NEDDylation but also ubiquitination, in the case of NUB1L/NUB1 through the presence of UBA domains and *en-bloc* degradation of conjugates. In previous studies, we observed a complex effect of NUB1 on protein ubiquitination (multi-mono vs poly) (Liu et al., 2010), which we feel complicates the interpretation of results.

As discussed above, we believe that the most appropriate experimental approach is to manipulate the levels of NEDD8 under conditions where CRL activity is not affected.

The addition of the new data in Fig. S6 where we determine the differences in the half-life of RPL7 depending on its modification either by UBA1-dependent NEDD8 or ubiquitin, further supports the presented hypothesis that NEDD8 compromises substrate targeting and proteasomal degradation.

Reviewer #2 (Remarks to the Author):

In the manuscript entitled, "Atypical NEDDylation promotes nuclear protein aggregation and protects the Ubiquitin Proteasome System upon proteotoxic stress," Maghames and colleagues present evidence for atypical NEDDylation of ribosomal and nuclear proteins and present arguments supporting that NEDDylation of substrates in a stress dependent manner is a defense mechanism against proteotoxic stress in the cell. They also identify HUWE1 (a E3 ligase) as an integral component for atypical NEDDylation. The results provided in this manuscript claim that concomitant aggregate formation during UPS dysfunction is not only the cause of proteotoxicity but is also a defense mechanism against the same and NEDDylation is the key towards counteracting this stress.

NEDD8 mediated substrate modification is analogous to ubiquitin modification of proteins which is an essential process during protein quality control and turnover. NEDD8 modification of proteins is carried out via both a canonical (NAE, Ubc12, Ube2F) pathway or via an atypical pathway requiring Ube1 instead. Various stressors such as heat shock, proteasome inhibitors as well as oxidative stress have been shown to increase protein NEDDylation however there is less evidence for biological significance of atypical modifications. It has been suggested that NEDDylation is involved in pathogenesis of various neurological disorders such as Alzheimer's disease essentially via driving the accumulated protein aggregates.

Overall the work is well-done and the experiments are presented well. However, there are major concerns that will need to be addressed to substantiate the central claims in this manuscript.

The main thrust of the manuscript is to characterize the biological significance of what the authors term atypical neddylation. This type of neddylation is distinct from the well-characterized canonical neddylation that uses the Nedd8 heterodimeric E1 enzyme to modify, almost exclusively, cullin substrates in a manner that modulates that activity of all cullin-RING ligases. So-called atypical neddylation occurs when the ubiquitin E1 enzyme, mistakenly charges Nedd8 instead of ubiquitin. This results in ubiquitin E2 enzymes accepting Nedd8 and then utilizing this charged Nedd8 species as they would with ubiquitin to catalyze transfer, in concert with ubiquitin E3 enzymes, to substrates. Essentially, "atypical neddylation" is analogous to ubiquitination except that Nedd8 is used in place of ubiquitin. This has been shown to occur in cells in conditions that lower the concentration of free ubiquitin such that the relative pools of free ubiquitin and Nedd8 now allow for mis-charging of Nedd8 by the ubiquitin E1 enzyme because there is little free ubiquitin to charge. The ability of the ubiquitin E1 to utilize Nedd8 has been well-established. Here the authors use heat-shock which, like proteasome inhibition, results in a lowering of free ubiquitin levels that allow for an increase in atypical neddylation. What the biological significance, if any, of this atypical neddylation is not well established.

Response

The key argument of the reviewer is that proteotoxic stress, by lowering the concentration of ubiquitin in cells, allows the mis-activation of NEDD8 by the ubiquitin system, ie NEDD8 behaves as ubiquitin with no specificity.

This is a valid hypothesis but multiple data including the new data in our manuscript support an alternative model:

1. Despite the close sequence similarity between NEDD8 and ubiquitin, thermodynamic studies showed that NEDD8 is thermodynamically much more unstable compared to ubiquitin (Kitahara et al., 2006). This difference may be particularly important during proteotoxic stress when NEDD8 is incorporated in hybrid NEDD8/ubiquitin chains.
2. The hypothesis that the activation of NEDD8 by the ubiquitin enzymes is due to the decreased levels of free ubiquitin has a direct prediction: Overexpression of ubiquitin should reverse the effect and block the activation of NEDD8 by the ubiquitin pathway. Even if we overexpressed ubiquitin at levels exceeding multiple times the endogenous levels, the NEDD8 response to stress remains unaffected (data available upon request). While we do believe that the depletion of free ubiquitin participates in the NEDD8 response to stress, it may not be the major element.
3. The data on HUWE1 show a high specificity for the NEDD8 response to stress and suggest that NEDD8 does not simply replace ubiquitin (please also see below).

The alternative hypothesis supported by the presented study is that stress-induced NEDDylation impacts of nuclear protein aggregation and UPS function during proteotoxic stress.

Major issues:

1) The tools to differentiate between canonical neddylation and atypical neddylation in this study are utilization of the Nedd8 E1 inhibitor MLN4924 to inhibit canonical neddylation and siRNA-mediated knockdown of Nedd8 to inhibit atypical neddylation. While it is extremely clear that MLN4924 treatment inhibits canonical neddylation it is less clear that knockdown of Nedd8 only impacts atypical neddylation. In fact, no evidence is presented that knockdown of Nedd8, which one would predict would impact ALL neddylation, only inhibits atypical neddylation. The authors make an argument that knockdown of nedd8 only inhibits atypical neddylation in data presented in figure 2D. They argue that knockdown of nedd8 reduces the amount of high molecular weight neddylated species that are induced upon heat shock (which is certainly true) but does not reduce canonical neddylation. The second claim is not substantiated by the data as there is a clear reduction in the amount of cullin neddylation (dark band just above 100kD) upon knockdown of Nedd8 in both untreated and heat-shocked conditions. As such, knockdown of Nedd8 is effecting BOTH canonical neddylation and atypical neddylation making any claims about the function of atypical neddylation using knockdown of Nedd8 impossible to interpret. How do we know that any of the effects on aggregation or turnover of reporter proteins (NLSGFPu) arise from inhibition of canonical neddylation. There are certainly hints that there are real differences here. This is best represented by figures 2F-H where knockdown of nedd8 appears to increase protein "aggregation" whereas MLN4924 treatment has a much different effect. There appears to be some evidence that mixed Nedd8-Ub chains may alter the solubility of proteins, but MUCH more biochemical evidence is needed to

support this claim, especially due to the ambiguity of the Nedd8 knockdown approach.

Response

All major experiments on NEDD8 knockdown are compared to MLN4924 treatment, which completely blocks NAE dependent NEDDylation but not stress-induced NEDDylation. This comparison should identify and exclude any effects of NEDD8 knockdown on the “canonical” NAE-dependent NEDDylation.

Please refer to the full response for reviewer 1, point 3.

2) The authors routinely utilize a biochemical separate technique to differentiate soluble from aggregated protein. While this is a good idea, the technique that the authors use comingles “aggregated” proteins with chromatin. It is clear that there is chromatin in the pellet fraction (Figure 1C) as well as ubiquitinated and neddylated protein. As such, this fraction likely contains other proteins that are not aggregated so any claims that proteins identified in this fraction by mass spectrometry are aggregated or less-soluble than are not substantiated by the data.

Response

All fractionation experiments including those for mass spectrometry were performed using established protocols that are routinely used in the field for the analysis of proteotoxic stress-induced aggregates. Typically, these protocols include high concentrations of detergent and/or sonication to indeed remove chromatin interacting proteins from the isolated insoluble pellet. While it is impossible to exclude that some chromatin-related proteins are present, we think the vast majority indeed represent insoluble proteins. This issue is not specific to the presented study.

3) The significance of the Ring-like nuclear structures are not well-established. Certainly, nedd8 and ubiquitin localize to these structures that seem to encircle RPL7-GFP, but it is not clear if this is some sort of nuclear quality control compartment. Are other ribosomal proteins present within the Ring? Are other cytoplasmic proteins found to be de-enriched in pellet fractions upon Nedd8 knockdown found in those rings? Are other well-characterized nuclear aggregating proteins found in these rings? Is the formation of these rings upon heat-shock blocked by inhibition of the ubiquitin E1 or the Nedd8 E1?

Response

As mentioned in the Discussion, similar to the presented nuclear structures were reported in Latonen et al., 2011, where by immunostaining multiple cell cycle proteins were shown to localize. We also show that in addition to RPL7, RPL11 is localized within these nuclear structures. We now show that the NEDD8 stained nuclear aggregates depend on UBA1 but not NAE E1 enzyme a key characteristic of the so-called “atypical” NEDDylation.

We would like to note that based on the presented 3D re-construction images we removed the term “ring-like” (please see below).

4) The SILAC experiments are very hard to interpret. The authors enrich for pellet fraction proteins upon heat shock and find nearly every highly abundant protein has a log2 ratio > 1. This result is not surprisingly in the least, as heat shock will drive protein

aggregation and nearly every protein will be found in this pellet fraction. In fact, their distribution shown in Figure 2B is entirely skewed to the right. The authors chose to highlight ribosomal proteins in this plot for unknown reasons as almost every highly abundant protein can be found in this list, Tubulin, proteasome components, metabolic enzymes, etc. The choice to highlight ribosomal proteins is not well-established and seems random as there many proteins whose log2 ratio is >1 in this experiment. The exact same statement can be made about the data presented in figure 2F. Almost every protein identified shows less presence in pellet fractions upon nedd8 knockdown. This result is very non-specific (again as the data is entirely skewed), so why focus on nuclear proteins or ribosomal proteins? Do ribosomal proteins even pass a significance test here given that the entire population is skewed to the left? They seem to be in the bulk of the population? What proteins are 2 or 3 standard deviations away from this skewed mean?

Response

The observed shift of proteins to the right upon heat shock is not surprising as in unstressed conditions there is minimal aggregation. This SILAC experiment specifically addresses protein abundance in aggregates and not in total cell extracts where a normal distribution is expected.

We agree that multiple groups of proteins are affected upon heat shock. However, we hope the reviewer will acknowledge that it is impossible to validate and follow all identified targets.

The selection of ribosomal proteins and subsequent analysis was for the following reasons:

1. The analysis of proteomic data presented in Fig 2C, shows that ribosomal proteins represent one of the top group of proteins within the heat shocked induced aggregates with one of the highest confidence (FDR).
2. The comparison of the proteomic data for siNEDD8 and MLN4924 in Fig. 2G identified ribosomal proteins as the top group of proteins, which are specifically affected by siNEDD8 and not by inhibition of the NAE dependent NEDDylation. This was critical to define potential targets for the stress induced NEDDylation.
3. Recent studies suggest a role for ribosomal proteins in the proteotoxic stress response.
4. Ribosomal proteins have been reported as NEDD8 substrates, providing good model substrates for subsequent studies.
5. Ribosomal proteins were recently identified as specific substrates for HUWE1 during proteotoxic stress, the E3-ligase that is identified and characterised in our study as the key E3-ligase for stress-induced NEDDylation.

As a general principle, it is accepted that proteotoxicity is more likely to be defined and controlled by a group of proteins rather than by individual proteins. This was yet another reason, for which we turned our attention to RPs and not to individual proteins. We have now included the above arguments into the text to clarify the choice of ribosomal proteins for subsequent analysis.

In addition, protein aggregation by definition is characterized by complex formation of proteins. It is very likely that modification of proteins with ubiquitin/ubls allows the recruitment of other proteins of the same or different family indirectly into aggregation. Thus, in the case of the NEDD8 knockdown experiment many proteins maybe “released”

from the aggregates indirectly, due to lack of interaction with NEDDylated proteins. It was thus critical to identify and characterize direct NEDD8 substrates (such as RPL7) present in the aggregates, as these proteins may act as “seed” molecules for aggregation. The tables in Fig. 2C, 2G represent the group of proteins and biological processes affected by heat shock and specifically by siNEDD8. We feel the tables provide the key information for the readers, who could potentially test the role of NEDD8 in proteotoxic stress response through additional targets/pathways.

5) The authors utilize cytoplasmic or nuclear localized version of unstable GFP (GFPu) to attempt to show that atypical neddylation is required for the efficient turnover of nuclear ubiquitin-proteasome substrates. They clearly show that NLSGFPu but not NESGFPu accumulates upon heat shock. The authors don't comment on the reason for this difference. Both substrates would utilize the exact same enzymes to catalyze their turnover as they have the same degron, so why would they behave differently upon heat shock? No explanation is given.

Response

In the results, we mention that this difference in UPS sensitivity is potentially due to differences in the relative abundance of UPS in the cytoplasm vs nucleus and/or lack of auxiliary proteolytic systems in the nucleus, such as autophagy. However, the important information from this experiments is that both heat shock and NEDD8 inhibition, control the nuclear but no cytoplasmic UPS. This shows specificity of the response and further supports the argument that NEDD8 is a regulatory pathway for nuclear UPS function during proteotoxic stress.

Further, the authors claim that the accumulation of NLSGFPu upon heat shock is exacerbated upon Nedd8 knockdown and this is entirely due to some mechanism that relies on atypical neddylation. However, this is based on small differences between Nedd8 knockdown and control knockdown (24.1 fold vs 29.8). These kinds of differences can arise from differences in western blotting which has limited quantitative resolution (especially if the authors are using film for their immunoblots). I would want to see true biological replicate experiments showing a reliable and truly quantitative difference between NLSGFPu levels upon heat shock for this result to be meaningful.

Response

We feel that the 24.1 vs 29.8 (180min) is not the correct comparison. It is clear that the response, especially upon NEDD8 knockdown, is saturated at late time points (180min). At earlier time points, for example 45 and 90min, the difference between control and siNEDD8 is quite clear (1.8 vs 16.4) and (14.9 vs 28) respectively. We now present quantitation of 3 independent experiments showing the reproducibility of the experiment. We would like to note that the effects are specific to NEDD8 knockdown, not observed upon inhibition of the NAE dependent NEDDylation.

6) The authors seem to be making an argument that Neddylation drives proteins toward aggregation and using experiments like those depicted in figure 4 to make that argument. While it is clear that there is more neddylated RPL7 in the pellet upon heat shock, is this specific for nedd8 or RPL7? What if the authors repeated this experiment using His-Ubiquitin. Would the result be different? What about for any of the hundreds of other

proteins whose presence in the pellet fraction increases upon heat shock (like tubulin, or GAPDH?). If this is just mimicking ubiquitination, then atypical neddylation is merely a biochemical artifact.

Response

We would like to note that the NEDD8 overexpression experiment was used as an alternative approach to assess the role of NEDD8 in protein aggregation in the absence of any stress. We tested the overexpression of ubiquitin, which in contrast to NEDD8 does not appear in the insoluble pellet. In addition, in the fractionation experiment performed for NEDD8 we blotted for nucleolin. We selected this candidate as the proteomic analysis indicates that heat shock increases the abundance of nucleolin in the insoluble pellet, but independently of NEDD8. This was confirmed by the western blot analysis, suggesting that NEDD8 does not promote unspecifically nuclear protein aggregation during the heat shock response. The identification of HUWE1 as a specific E3-ligase for stress-induced NEDDylation further supports the above notion.

7) The idea that atypical neddylation drives proteins to aggregate and thus protect the ubiquitin proteasome system is complicated by the authors own data. In figure 4G, the authors overexpress Nedd8, which, according to the authors hypothesis, should drive proteins toward aggregation. However, there is clearly less RPL7 in the pellet fraction upon Nedd8 overexpression. This argues that atypical neddylation prevents aggregation of at least RPL7.

Response

We are not sure how the above conclusion was derived. In Fig. 4H the correct comparison is Soluble/Pellet of the pcDNA3 untreated lanes (first two) with the Soluble/Pellet ND8 overexpression lanes (last two). This clearly shows that NEDD8 overexpression promotes aggregation of RPL7 consistent with the presented hypothesis. Heat shock (middle two lanes) causes the highest increase of RPL7 in the insoluble pellet, but this is expected, as heat shock in addition to post-translational modifications induces protein unfolding, which is the key contributor of aggregation.

8) All the data with Huwe1 merely represents a biochemical phenomenon that takes place upon lowering free ubiquitin levels (like upon heat shock). Because the ubiquitin E1 is mistakenly utilizing Nedd8 instead of ubiquitin, many, if not all, ubiquitin E2 enzymes will accept this activated Nedd8 from the Ubiquitin E1 enzyme and then utilize this Nedd8 as it would ubiquitin in all transfer reactions with E3 enzymes, like Huwe1. Thus, ANY ubiquitin ligase would show increased transfer of Nedd8 to its substrates under these conditions, including Huwe1. The authors are just merely demonstrated a biochemical aberration that occurs upon heat shock that could be demonstrate for any ubiquitin ligase/substrate pair.

Response

The proposed hypothesis by the reviewer is that NEDD8 behaves as ubiquitin and any E2 could accept NEDD8 and through any E3-ligase substrates are modified. This hypothesis however has a direct prediction: That knockdown or knockout of a single E3 will not affect global ubiquitination/NEDDylation as it will be compensated by the action of the multiple present E3s. Indeed, this is the case for ubiquitination. Knockdown of HUWE1 has no significant effect on global ubiquitination (Fig 5C). In direct contrast under the

same conditions stress-induced NEDDylation is dramatically reduced. We believe these data are critical as they:

1. Identify HUWE1 as the first E3-ligase that specifically promotes stress-induced but not “canonical” (NAE dependent) NEDDylation.
2. Demonstrate that the stress-induced NEDDylation is a rather specific response and that NEDD8 *in vivo* is not conjugated by any E3 ligase to substrates as ubiquitin. The data suggest that HUWE1 is the main if not the only E3 responsible for stress-induced NEDDylation. This is also consistent with the idea that ribosomal proteins could represent a major target for stress-induced NEDDylation, as HUWE1 was recently identified as specific E3 for ribosomal protein ubiquitination.

We will also like to mention that we have tested additional E3-ligases, including NEDD4 and MDM2, which have no effect on stress-induced NEDDylation (data available upon request), further supporting the HUWE1 specificity in the response.

9) In order for the authors to argue that atypical neddylation alters the solubility and turnover of proteins in which atypical neddylation occurs, the authors need to actually demonstrate either of those outcomes. For instance, does the rate of turnover, using metabolic pulse chase assays, of RPL7 or some collection of putative atypical neddylation substrates, change, at steady-state, and upon heat shock upon nedd8 knockdown.

Response

Heat shock compromises the UPS function (Fig. 3) and therefore it is not possible to gain meaningful data on the half-life of a protein under conditions where the UPS is not functioning. We however performed an alternative experiment, which we believe addresses the reviewer’s comment:

We now show that under conditions of NEDD8 overexpression RPL7 is NEDDylated through UBA1 and not NAE (ie. “atypically”) (Fig. S6). We thus measured the half-life of RPL7 modified either by “atypically” activated NEDD8 or by ubiquitin. The data show that “atypically” NEDDylated RPL7 is much more stable compared to ubiquitinated RPL7. This is consistent with the presented hypothesis that NEDD8 compromises substrate degradation.

Even this result would be a bit inconclusive due to the inherent problems in nedd8 knockdown (see point 1). Further, does a protein with a mixed nedd8-ubiquitin chain get destroyed by the proteasome at a different rate than the same protein with a pure ubiquitin chain (of the same length). This kind of biochemical demonstration of a difference between a mixed nedd8-ub chain and a pure ub-chain would be required to begin to clearly demonstrate a role for these mixed-chains (which clearly can form in cells) in regulating protein turnover.

Response

Such experiments require recombinant HUWE1, which based on our studies, is the key E3 ligase that promotes “atypical” NEDDylation. HUWE1 is approximately a 500kDa protein and not possible to recombinantly express it. Its HECT domain, which is possible to express, does not interact with the substrates. In addition, we are not aware of an established biochemical approach that can consistently ensure modification of a substrate

in vitro with a defined length of ubiquitin chains. The above issues do not currently allow us to perform such experiments. However, we believe the experiment performed in S6 (see above) provides insights on the role of “atypical” NEDDylation on 26S proteasomal degradation of substrates.

Minor points:

The immunofluorescence microscopy in Figure 1E with DAPI staining is hardly visible and most of the microscopy should show a gray scaled version of each panels. A 3D surface rendering would be a good way of showing effective co-localization. siRNA treatment of NEDD8 coupled with immunofluorescence microscopy might also be a good option.

Response

We now present all immunofluorescence analysis in gray scale, with the merge in colour. Indeed, the 3D rendering experiment was informative as it revealed that the “ring-like” structures originally observed on a single z stack are part of a nuclear sphere-like structure, which is decorated on the surface with NEDD8/ubiquitin.

We now include immunofluorescence experiments upon NEDD8 knockdown, NAE and UBA1 inhibition in heat shocked and MG132 treated cells (Fig. 1F, S1D). The data support the notion that the NEDD8 conjugates found in nuclear aggregates depend on UBA1 but not NAE enzyme, a key characteristic of “atypical” NEDDylation.

Demonstrating NEDD8 co-localization with RPL7-GFP. Showing similar data with another ribosomal protein such as RPL8 or RPL11 should strengthen the argument.

Response

We include immunofluorescence experiments for RPL11-GFP, which shows very similar patterns to RPL7-GFP upon heat shock and MG132 (S4B).

Page 5 – Typo – “This study” instead of “The studies”

Response

Corrected.

Figure 4B, C, E – Inset panel points to MG132 treatment and needs to be labeled.

Response

Corrected.

Pearson’s Coefficient calculate in Figure 1 is not described in the methods section.

Response

We now describe in the methods (Immunofluorescence microscopy) the calculation of Pearson’s Coefficient.

How was the FDR calculated in Figure S2?

Response

We now describe in the methods (SILAC-Mass Spectrometric Analysis) the calculation of FDR.

How was the relative GFP level calculated in Figure 3? ImageJ? Was film used?

Response

For all western blot quantifications, film and ImageJ were used. We describe this in the methods (Western blot analysis).

Reviewer #3 (Remarks to the Author):

This manuscript reports the atypical neddylation of newly synthesized proteins that are either misfolded or form aggregates during the heat shock treatment. Using metabolic labeling (SILAC), the authors observed an enrichment in NEDD8 in the insoluble pellet of cells that were treated to a heat shock, suggesting a role for protein neddylation in protein aggregation. Out of the ~1700 proteins quantified, they identified a subset of ~ 55 ribosomal proteins in the insoluble fraction upon heat shock treatment. By analyzing the turnover of NES- and NLS-GFP constructs they determined that the nuclear proteasome is impaired during the heat shock response, an effect that is accentuated by knocking down the NEDD8 machinery. Follow-up experiments on RPL7 confirmed the neddylation of this substrate during heat stress and its increased aggregation when Nedd8 is overexpressed. Immunofluorescence microscopy experiments confirmed that HECT E3 ligase HUWE1 colocalized in heat shock-induced aggregates, and knock down of this ligase also reduced neddylation of RPL7, thus suggesting a role for HUWE1 in atypical protein neddylation. By knocking down NEDD8, the authors found an increased ubiquitination of RPL7 indicating that neddylation competes with ubiquitination during heat shock. Altogether, these results suggest that atypical NEDD8 conjugation may protect substrates from proteasome degradation during heat shock by favoring protein aggregation.

Overall, the manuscript provides valuable information on the potential interplay between neddylation and ubiquitination during heat stress. The authors provide appropriate data to support their claims, though several experiments lack replicate to evaluate the statistical significance of their findings. For example, all SILAC experiments are conducted on single injection with no replicate, and more than 10% of abundance measurements are obtained for protein quantified with only one peptide. Other reproducibility measurements should be provided for immunofluorescence microscopy experiments. Also, there is an overwhelming number of figure panels (Figures 1 and 4), and the authors should make an effort to move non-essential display items to supplementary material. Additional points are outlined below:

1. Figure 1, panel H) should be moved to the supplemental. For Panel E) there should be a bar graph to show that these ring formations are statistically significant in the MG132 treated cells. Error bars should be provided for panel G).

Response

Based on the reviewer's suggestion and the addition of new data we have now removed or transferred many of the data to supplementary information.

We also provide statistical analysis (with error bars) for all performed immunofluorescence analysis from 3 independent experiments for each condition. Approximately 100 cells were used for each condition/experiment.

2. In Figure 2B and the accompanying text, it is surprising that the authors do not comment on the interplay with other UBLs such as SUMO. Interestingly, their supplementary table reports the occurrence of all three SUMO paralogs in aggregates, raising the possibility that protein sumoylation may also contribute as previously

reported for heat shock treatment (e.g. *Sci Signal.* 2009 May 26;2(72):ra24; *Cell Div.* 2015 Jun 20;10:4). A comment regarding the significance of protein modification by other UBLs is warranted.

Response

We agree with the reviewer and now include a paragraph reporting the identification of SUMO-1, 2 in our heat shock experiments, consistent with previous studies. In addition, in Fig. 2B we indicate the presence of SUMO-1, 2 in the scatter plot of the HS experiment. SUMO-3 was quantified in one of the 2 experiments (increased aggregation) and is not presented. The following references were included: Golebiowski, F. *et al.*, 2009 ; Enserink, J. M, 2015 ; Hendriks, I. A. & Vertegaal, A. C, 2016.

3. Replicate SILAC experiments should be reported for data shown in Figures 2 B), F), G), and H) to determine the statistical significance of abundance changes measured.

Response

For all performed SILAC experiment we present the mean of 2 replicate experiments and the Pearson correlation coefficient in Fig. 2, S2. Proteins with at least 2 peptides, quantified in both experiments are presented.

4. The amount of cell extracts used for immunoblots in the input and after NTA purification should be reported.

Response

We now report the amount of cell extract used for input (Western blot analysis) and Ni-NTA purification (Ni-NTA pull-down of NEDDylated substrates from the pellet fraction) in the methods.

5. For Figure 4, why is ubiquitin found in the nucleus of MG132-treated cells in panel C) and in the cytoplasm in panel E)?

Response

In general, we found that prolonged MG132 treatment causes the formation of ubiquitin stained nuclear aggregates with reduction in the nucleoplasmic ubiquitin staining. We think this is due to sequestration of nuclear ubiquitin into the nuclear aggregates upon prolonged proteotoxic stress. This sequestration is not observed upon short-term heat shock. The only exception is the experiments performed with RPs-GFP constructs where the depletion of nucleoplasmic ubiquitin is not observed upon MG132 treatment and the ubiquitin staining in the aggregates is weaker. We do not have an explanation for this, but it may be due to GFP expression, as it has been reported that GFP can interfere to some extent with protein ubiquitination. However, we do not feel that this affects (if anything it underestimates) the conclusion of these experiments.

6. For Figure 4 G), in the right bottom panel, why is the pelleted RPL7 not conjugated by Ubi or NEDD8? The mass shown on the blot corresponds to the unmodified form. Why is the signal so faint compared to the input?

Response

In principle, as only a small % of the substrate is modified with Ub/UBLs it is not always possible to detect the modified forms in total input of aggregated proteins, especially under short-term heat shock conditions (presented in this experiment). As shown in Fig. 4

it is important to isolate these conjugates from the aggregate pellet for a clear detection. We had presented a low exposure of the blot in Fig. 4G (now 4H) to clearly show the difference in RPL7 aggregation in control and NEDD8 transfected conditions. We have now included a high-exposure, where it is possible to detect a weak signal for the modified RPL7 in the pellet upon heat shock.

7. *For Figure 5 C), is there a less exposed version of the blot that would allow to quantitate the conjugated cullin?*

Response

We now include a low exposure for the analysis of extracts in the experiment presented in Fig. 5C (Fig. S5). Occasionally, we do observe a small increase in cullin NEDDylation upon MG132 treatment but it is independent of HUWE1. Regarding the effect of minor changes in cullin NEDDylation on CRL activity, please refer to the response for Reviewer 1, comment 3.

8. *Figure 6 C), in the right panel, why is the RPL7 not modified by NEDD8 or Ubiquitin (no mass shift by gel). Could it be that RPL7 is not actually neddylated, and that but another protein from the ribosomal complex is?*

Response

The lysis and immunoprecipitation experiments are performed under native conditions in the absence of any protease inhibitors. This is required as we aim to isolate active proteasomes. However, this has the consequence that de-conjugating enzymes, including the ones present on the proteasome, most likely, deconjugate the modified forms of RPL7 during experimentation. We extensively discussed this issue with experts in the proteasome field who share similar views.

We feel that we provide strong evidence for the modification of RPL7 with NEDD8. However, the proposed hypothesis that non-modified substrates can be driven to the proteasome through a complex formation with modified targets is clearly valid. We now include this possibility in the results section.

9. *For Figure 6 B), right panel, the membrane was blotted with what antibody? Is it HIS or HA?*

Response

We used anti-HA antibody for the experiment and it is now mentioned in the figure.

10. *Check text for inconsistency and typos (Line 482 “Humanentries” should be “Human entries”)*

Response

Corrected.

Reviewers' comments:

Reviewer #1 (Remarks to the Author):

In this revision, the authors provided new data to support that 1) NEDDylation inhibits RPL7 degradation and 2) HUWE1 participates in stress-induced NEDDylation. While these observations provide a potential regulatory mechanism for stress-induced NEDDylation, it is less clear with regard to the biological consequence of such NEDDylation (nuclear protein quality control) and its physiological/pathological relevance.

Reviewer #2 (Remarks to the Author):

The authors have not successfully addressed my previous concerns. Chief among them is the problem with using siNedd8 to discriminate canonical vs atypical neddylation. Their response is just another argument that I should "believe" their results when their own data argues otherwise. Further, their mass spec data suggest that a large fraction of highly abundant proteins are driven to aggregates upon heat shock and this is altered upon siNedd8. They attempt to argue some specificity by examining ribosomal proteins when it appears that there is very little specificity. Their response to this query is unsatisfactory and is their response to other concerns.

Reviewer #3 (Remarks to the Author):

I am satisfied with the response of the authors who answered most of the queries I raised. I have no further comments and recommend publication of the revised manuscript in its current form.

Reviewer #1 (Remarks to the Author):

In this revision, the authors provided new data to support that 1) NEDDylation inhibits RPL7 degradation and 2) HUWE1 participates in stress-induced NEDDylation. While these observations provide a potential regulatory mechanism for stress-induced NEDDylation, it is less clear with regard to the biological consequence of such NEDDylation (nuclear protein quality control) and its physiological/pathological relevance.

Based on the previous comments by the reviewer we followed 2 additional approaches to strengthen the conclusions regarding the role of stress-induced NEDDylation:

1. We used the overexpression of NUB1 as a tool to repress atypical NEDDylation. We found that NUB1 overexpression:
 - a. Represses the formation of stress-induced NEDD8 conjugates found in aggregates and prevents stress-induced RPL7 aggregation (Fig. 4I).
 - b. Reduces NEDDylation of RPL7, while it promotes RPL7 ubiquitination (Fig. S6).
 - c. Promotes the interaction of RPL7 with the proteasome (Fig. 6D).

The data are consistent with the presented role of NEDD8 in protein aggregation upon stress and strengthen the conclusions made with the use of siNEDD8.

We want to emphasise that our previous studies on the role of endogenous NUB1 (knockdown experiments) on substrate ubiquitination and atypical NEDDylation showed a complex effect between mono- vs poly-ubiquitination. Thus, we do not make any claims on the physiological role of NUB1 on the process. We only use NUB1 overexpression as a complementary approach to siNEDD8.

2. We performed *in vitro* assays where we monitor the efficiency of purified 26S proteasomes to process/degrade either ubiquitin or hybrid NEDD8/ubiquitin conjugates isolated from cells. The data show that in contrast to ubiquitin, hybrid NEDD8/ubiquitin conjugates are resistant to 26S proteasome processing (Fig. 6F). The data strengthen the previous presented data showing that NEDD8 blocks degradation of RPL7 and provide a biochemical explanation on how NEDD8 promotes protein aggregation during stress; by preventing processing/degradation of substrates by the proteasome.

Reviewer #2 (Remarks to the Author):

In general, we find the reviewer's comments harsh and unfair. We had provided a quite extensive and detailed response to the reviewer's concerns based on scientific evidence presented either in our manuscript or in previous studies.

Specific points:

The authors have not successfully addressed my previous concerns. Chief among them is the problem with using siNedd8 to discriminate canonical vs atypical neddylation.

It is not the siNEDD8 approach that discriminates canonical vs atypical NEDDylation but rather the combination of siNEDD8 with the MLN4924 treatment. MLN4924 is currently the only tool that specifically blocks canonical but not atypical

NEDDylation. The comparison of the siNEDD8 data with those upon MLN4924 treatment indicates the effects of siNEDD8 that are not due to the inhibition of canonical NEDDylation. Currently, the only additional known function of NEDD8, especially under proteotoxic stress, is the atypical. A well-characterised tool that exclusively blocks atypical NEDDylation does not exist.

In addition to the siNEDD8 approach, we used: inhibitors of UBA1 and NAE, expression of NUB1 (new data), knockdown of HUWE1, half-life experiments on atypical NEDDylated proteins, *in vitro* 26S proteasome assays (new data). It is the combination of all these data that support the role of atypical NEDDylation.

However, and as noted in our previous response we only demonstrate the role of NEDD8 in the proteotoxic stress response. In the discussion section we mention the potential role of atypical NEDDylation, based on the presented data.

Their response is just another argument that I should "believe" their results when their own data argues otherwise.

No description of the data that argue against our presented argument(s).

Further, their mass spec data suggest that a large fraction of highly abundant proteins are driven to aggregates upon heat shock and this is altered upon siNedd8.

We don't understand what is the issue here. We are not the first to determine the composition of proteotoxic stress induced aggregates. Previous studies in *S. cerevisiae* and *C. elegans*, showed the aggregation of highly abundant proteins upon stress. Indeed, in our experiments the aggregation of abundant proteins is affected both by siNEDD8 and MLN4924 treatment (we assume the ones that the reviewer refers to). However, such proteins were excluded from subsequent analysis. We bioinformatically analysed only the proteins that were affected by siNEDD8 but not by MLN4924 (to exclude canonical NEDDylation). This analysis identified ribosomal proteins as the top group of proteins with the highest confidence score. However, it is expected that siNEDD8 (or any treatment) will indirectly control the aggregation of other proteins (please see below). Thus, we do not claim specificity at this level.

They attempt to argue some specificity by examining ribosomal proteins when it appears that there is very little specificity. Their response to this query is unsatisfactory and is their response to other concerns.

We assume that the lack of specificity refers to the general effect of siNEDD8 and/or MLN4924 on aggregate composition. These treatments have direct and indirect effects. For example, this has been recently shown for the role of NEDD8 in stress granule assembly. Inhibition of NEDDylation fully blocks the formation of stress granules induced upon oxidative stress, including many abundant proteins (Jayabalan et al., 2016, Nat.Com.). Where is the specificity of NEDD8 in this response? Clearly, not all stress granule proteins are NEDDylated but instead NEDD8 controls stress granule assembly through modification of few proteins (for example SRSF3).

The comparison between the siNEDD8/MLN4924 proteomics data identified ribosomal proteins as the top group of proteins affected by siNEDD8 but not by MLN4924, with the highest confidence score, ie as potential substrates for atypical NEDDylation. To reveal the direct effects of NEDDylation on nuclear aggregation, all

subsequent experiments to the proteomics analysis were focused on the validation and characterisation of ribosomal proteins as substrates of atypical NEDDylation. Namely:

1. The characterisation of RPL7 as direct model substrate of atypical NEDDylation upon proteotoxic stress
2. The localization of RPL7/RPL11 in stress-induced NEDD8 aggregates
3. Identification of the E3 that atypically NEDDylates RPL7
4. Characterization of the role of NEDD8 in RPL7 proteasome targeting and degradation.

Is this not a common and logic strategy of a follow-up study on proteomics data?

It is expected that decrease in NEDDylation will indirectly affect the aggregation of other non-modified targets-similarly to stress granules assembly (see above). Thus, we do not claim specificity of NEDD8 on aggregate formation-It may not simply exist. We also do not claim that ribosomal proteins are the only atypically NEDDylated targets. We use ribosomal proteins as model substrates in our study.

The specificity we refer to is at the level of the NEDD8 response to stress and substrate modification. The key argument of the reviewer in the initial report was that there is no specificity in the response and NEDD8 behaves as ubiquitin and any E3 will NEDDylate any ubiquitin substrate: *“Because the ubiquitin E1 is mistakenly utilizing Nedd8 instead of ubiquitin, many, if not all, ubiquitin E2 enzymes will accept this activated Nedd8 from the Ubiquitin E1 enzyme and then utilize this Nedd8 as it would ubiquitin in all transfer reactions with E3 enzymes, like Huwe1. Thus, ANY ubiquitin ligase would show increased transfer of Nedd8 to its substrates under these conditions, including Huwe1. The authors are just merely demonstrated a biochemical aberration that occurs upon heat shock that could be demonstrate for any ubiquitin ligase/substrate pair”*.

The presented data suggest that there is specificity in the NEDD8 response to stress. Based on our data, HUWE1 is the key if not the only E3 that promotes atypical NEDDylation. Thus, HUWE1 is the specificity factor for the NEDD8 response to proteotoxic stress. As recent studies identified ribosomal proteins as the main substrates of HUWE1 upon proteotoxic stress, it is reasonable to argue that ribosomal proteins may indeed represent the main (but not the only) substrates of atypical NEDDylation.

REVIEWERS' COMMENTS:

Reviewer #1 (Remarks to the Author):

I am satisfactory to the responses and do not have additional comments.